# SELF-GUIDED NOISE-FREE DATA GENERATION FOR EFFICIENT ZERO-SHOT LEARNING

**Jiahui Gao**[1]*, **Renjie Pi**[2]*, **Yong Lin**[2], **Hang Xu**[3], **Jiacheng Ye**[1],
**Zhiyong Wu**[4], **Weizhong Zhang**[5], **Xiaodan Liang**[6], **Zhenguo Li**[3], **Lingpeng Kong**[1]
[1]The University of Hong Kong [2]Hong Kong University of Science and Technology
[3]Huawei Noah's Ark Lab [4]Shanghai AI Lab [5]Fudan University [6]Sun Yat-sen University
{sumiler, carsonye}@connect.hku.hku,{rpi,ylindf}@connect.ust.hk,
lpk@cs.hku.hk,wuzhiyong@pjlab.org.cn,{xu.hang,li.zhenguo}@huawei.com,
{zhangweizhongzju,xdliang328}@gmail.com

## ABSTRACT

There is a rising interest in further exploring the zero-shot learning potential of large pre-trained language models (PLMs). A new paradigm called data-generation-based zero-shot learning has achieved impressive success. In this paradigm, the synthesized data from the PLM acts as the carrier of knowledge, which is used to train a task-specific model with orders of magnitude fewer parameters than the PLM, achieving both higher performance and efficiency than prompt-based zero-shot learning methods on PLMs. The main hurdle of this approach is that the synthesized data from PLM usually contains a significant portion of low-quality samples. Fitting on such data will greatly hamper the performance of the task-specific model, making it unreliable for deployment. Previous methods remedy this issue mainly by filtering synthetic data using heuristic metrics(e.g., output confidence), or refining the data with the help of a human expert, which comes with excessive manual tuning or expensive costs. In this paper, we propose a novel noise-robust re-weighting framework SUNGEN to automatically construct high-quality data for zero-shot classification problems. Our framework features the ability to learn the sample weights indicating data quality without requiring any human annotation. We theoretically and empirically verify the ability of our method to help construct good-quality synthetic datasets. Notably, SUNGEN-LSTM yields a 9.8% relative improvement than the baseline on average accuracy across eight different established text classification tasks.

## 1 INTRODUCTION

Owing to the superior generative capacity of large-scale pre-trained language models (PLMs), there has been an emerging trend of using these powerful models (e.g., GPT) to generate training data for downstream tasks (Anaby-Tavor et al., 2020; Puri et al., 2020; Kumar et al., 2020; Lee et al., 2021, *inter alia*). Among them, a new line of generation-based zero-shot learning using the unfinetuned PLM pushes the envelope further (Schick & Schütze, 2021; Ye et al., 2022a; Meng et al., 2022), featuring total annotation-free training for downstream tasks. Ye et al. (2022a) (ZEROGEN) further boosts the efficiency by using the generated data to train tiny task models (TAM), which have orders-of-magnitude fewer parameters than the PLM. Specifically, they first design prompts incorporating the task description and label information, then use them to guide the data generation from the PLM. Subsequently, the synthesized dataset is used to train the tiny task-specific models. Compared with the classic prompt-based zero-shot learning on PLM, this new paradigm enjoys two favorable properties: (1) since the task model has orders-of-magnitude fewer parameters than the PLM, it demonstrates much lower inference latency; (2) with the large amount of PLM-generated training data, the task model often shows better performance than prompt-based zero-shot PLM counterparts.

In the above paradigm, the amount and the quality of the generated data are crucial factors for the task model's performance. Unfortunately, despite the unlimited training data that one can generate

---

*Equal Contribution. Code is available at this link.

in theory, the data quality is not always guaranteed. Our experimental observation across many downstream tasks verifies the existence of this issue: in ZEROGEN, after a few training epochs on the PLM-generated dataset, although the training accuracy steadily improves, the actual test accuracy of the model starts declining rapidly (e.g., IMDb in Figure 1) – a clear indication of the model overfitting to low-quality data (noisy data) (Arpit et al., 2017). More specifically, we identify two major cases of noisy samples in the synthetic dataset: corrupted labels and task-irrelevant samples (Table 6 in Appendix). Without any task-related fine-tuning, it is challenging for PLM to follow a user's instruction (task-specific prompt including label information) to generate accurate samples in the target domain (Ouyang et al., 2022). To alleviate the data quality issue, recent work adopts human-active labeling to correct the corrupted label or revise the example (Wang et al., 2021a; Liu et al., 2022). However, such methods introduce considerable costs and may be unrealistic.

To avoid human intervention, the classic approach to eliminate the effect of noisy data is to re-weight the samples. The core idea is to design a weighting function $w$, such that the correct samples are associated with larger weights and the noisy ones with smaller weights. Compared with heuristic design of $w$ (e.g., according to output confidence, loss value) (Liu & Tao, 2015; Wang et al., 2021b), which requires task-specific knowledge and excessive manual tuning, the adaptive methods that learn the sample weights in an end-to-end manner demonstrate better performances in practice (Ren et al., 2018; Shu et al., 2019; Zheng et al., 2021). Those methods typically formulate the learning of sample weights into a bi-level optimization problem, with a clean validation set in the outer loop to

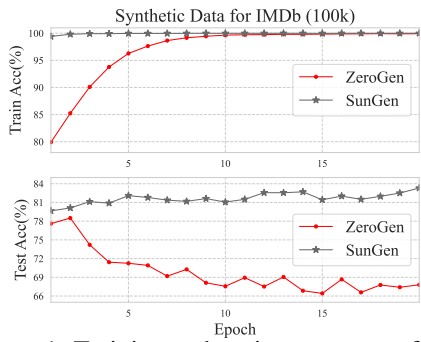

Figure 1: Training and testing accuracy of LSTM model trained on synthetic dataset. After training for more epochs, the testing performance of ZERO-GEN starts to deteriorate significantly, indicating that the model starts to fit the erroneous data.

guide the learning of $w$. Despite remarkable success was achieved, their dependence on a clean validation set becomes a major limitation, which is especially impractical in zero-shot setting.

Our solution comes from rethinking the choice of the outer objective in the bi-level framework: *can we design an objective such that the sample weights can be optimized with only access to the noisy synthetic data*? To this end, we resort to a family of noise-robust loss functions ($\ell_{robust}$) (Ghosh et al., 2017; Zhang & Sabuncu, 2018). These functions were adopted by previous work to train the neural network under label noise due to their theoretically noise-tolerant property (Ghosh et al., 2017; Zhang & Sabuncu, 2018; Wang et al., 2019). However, from the optimization point of view, such loss functions suffer from instability and difficulty when training the neural networks (Zhang & Sabuncu, 2018), which limits their effectiveness. Remarkably, our approach leverages the noise-tolerant property of these losses, while avoiding their pathology. We propose a novel bi-level re-weighting framework SUNGEN: in the inner loop, we train the task model using weighted training loss based on current sample weights; in the outer loop, the noise-robust loss is adopted to guide the learning of the sample weights. The two procedures are performed alternatively to generate a set of weights indicating the importance of samples. Notably, our method focuses on enhancing the quality of generated data, while improving the generator (e.g., modify PLM parameter, prompt engineering) is an orthogonal direction and can be applied jointly with our method.

Our main contributions are threefold. First, we propose a novel end-to-end framework to construct high-quality noise-free synthetic dataset, without the aid of any human annotation (§3). Second, we offer theoretical justification (§4) and empirical verification (§5.2) for SUNGEN's ability of recovering a noise-free dataset reliably with synthetic data only. Third, we conduct experiments on eight text classification datasets and show our method outperforms the current baseline by large margins (§5.2).

## 2 BACKGROUND

### 2.1 PROMPT-BASED ZERO-SHOT LEARNING

We first introduce prompt-based zero-shot prediction (named PROMPTING). Given a manually-designed prompt $\mathcal{T}(\cdot)$ and a query example $\mathbf{x}_i \in \mathcal{X}$, PROMPTING constructs a sentence $\mathcal{T}(\mathbf{x}_i)$ (e.g.,

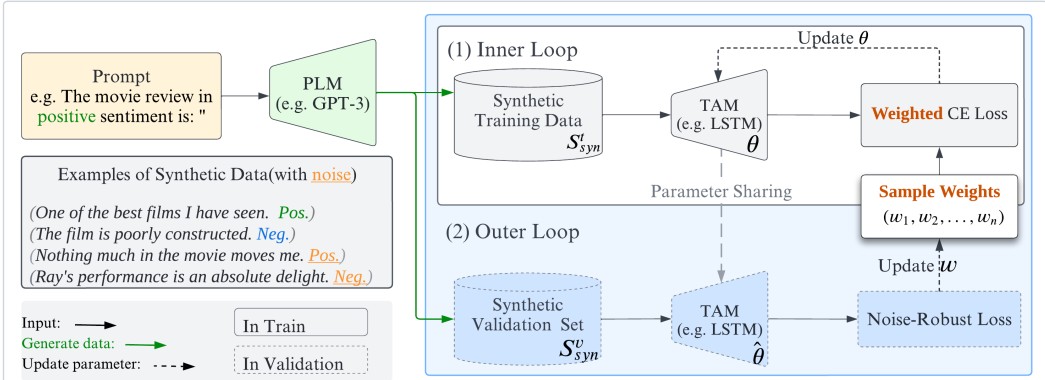

Figure 2: The framework of SUNGEN. Our bi-level framework learns sample weights $\boldsymbol{w}$ measuring data quality without relying on any human-annotated data. In the inner loop, we train a tiny task model (TAM) with weighted CE loss based on current sample weights, and produce trained TAM parameters $\hat{\boldsymbol{\theta}}(\boldsymbol{w})$; in the outer loop, we adopt a noise-robust loss to guide the learning of $\boldsymbol{w}$ by evaluating $\hat{\boldsymbol{\theta}}(\boldsymbol{w})$ on a synthetic validation set.

*"The movie review in <$y_i$> sentiment is: <$\mathbf{x}_i$> "*). The PLM $\mathcal{P}$ is expected to model the probability distribution of a set of label words $y_i \in \mathcal{Y}$ (e.g., "positive", "negative") and select the class with highest probability for $\mathbf{x}_i$. Even though PROMPTING has achieved remarkable success in zero-shot learning (Radford et al., 2019; Brown et al., 2020), it is difficult for PROMPTING to fully leverage the task-specific knowledge from PLMs. Besides, PROMPTING still needs to conduct inference on a cumbersome PLM.

## 2.2 EFFICIENT ZERO-SHOT LEARNING VIA DATA GENERATION

A new line of research (Ye et al., 2022a; Meng et al., 2022; Ye et al., 2022b) endeavors to make zero-shot learning more practical and efficient, among which the generative efficient zero-shot learning paradigm proposed by ZEROGEN (Ye et al., 2022a) is as follows:

**Synthetic Data Generation.** Given a task, the paradigm first generates a synthetic dataset $\mathcal{S}_{\text{syn}} = (\mathcal{X}_{\text{syn}}, \mathcal{Y}_{\text{syn}})$ with the help of large-scale PLM $\mathcal{P}$ and task-related prompts. The idea is to use model $\mathcal{P}$ to generate the input $\mathbf{x}_{\text{syn}}$ based on a pseudo label $y_{\text{syn}}$. For example, in text classification task, a class label $y_{\text{syn}}$ is uniformly sampled: $y_{\text{syn}} \sim \mathbf{U}(y_1, y_2, \ldots, y_K)$, where $K$ is the number of classes. The pseudo label $y_{\text{syn}}$ is transformed into a label-descriptive prompt $\mathcal{T}(y_{\text{syn}})$ to generate $\mathbf{x}_{\text{syn}}$: $\mathbf{x}_{\text{syn}} \sim \mathcal{P}(\cdot|\mathcal{T}(y_{\text{syn}}))$. The generated $\mathbf{x}_{\text{syn}}$ and pseudo label $y_{\text{syn}}$ are paired to construct a pseudo training dataset $\mathcal{S}_{\text{syn}}$.

**Efficient Training and Inference.** To achieve efficiently training and inference, the paradigm then trains a **t**iny t**a**sk **m**odel (TAM) (e.g., 1-layer Bi-LSTM) on the synthetic dataset $\mathcal{S}_{\text{syn}}$. TAMs typically have much fewer parameters, which is thus easy to train and more efficient during inference.

Although ZEROGEN achieved promising results by training a TAM with $\mathcal{S}_{\text{syn}}$, we find that model's performance rapidly declines after several training epochs, indicating overfitting to noisy samples (Figure 1). A classic approach to improve the quality of a dataset containing low-quality data is to re-weight the samples using some function $w$. Intuitively, if we assign large weights to correct samples and small weights to noisy ones, the negative influence of noisy samples during training can be reduced. However, the heuristic design of $w$ (e.g., according to output confidence, loss value) suffers unstable performances and requires task-specific knowledge (Shu et al., 2019). Therefore, our paper seeks to optimize $w$ automatically.

## 3 METHOD

To automatically learn sample weights, the bi-level optimization approaches (Ren et al., 2018; Shu et al., 2019) have been proven to be effective. However, such methods are impractical in our zero-shot scenario, since they depend on a clean validation set to guide the optimization of sample weights. To

circumvent the absence of human-labeled data, we propose a **S**elf-g**U**ided **N**oise-free data **GEN**eration framework, named **SunGen** (Figure 2). Concretely, we propose a sample reweighting framework via bilevel optimization using a noise-robust loss (Ghosh et al., 2017; Wang et al., 2019; Zhang & Sabuncu, 2018) as the outer objective. Due to the appealing property of such loss functions, our method is able to learn meaningful sample weights with only a synthetic (noisy) validation dataset.

**Notations.** As elaborated in Section 2.2, we first generate a synthetic dataset using a left-to-right PLM and task-related prompts for the given task. Let $\mathcal{S}_{\text{syn}}, \mathcal{S}_{\text{clean}} \subset \mathcal{X} \times \mathcal{Y} = \mathbb{R}^d \times \{1, ..., K\}$ denote distribution of synthetic (noisy) and clean (gold) data, respectively. Here $d$ is the dimension of the input space and $K$ is the number of classes. We draw a training dataset $\mathcal{S}_{\text{syn}}^{\text{t}} := \{(\boldsymbol{x}, y)_{(i)}\}_{i=1}^N$ and a validation dataset $\mathcal{S}_{\text{syn}}^{\text{v}} := \{(\boldsymbol{x}, y)_{(j)}\}_{j=1}^M$ from $\mathcal{S}_{\text{syn}}$. Denote $f(\boldsymbol{x}, \boldsymbol{\theta})$ as the classifier (TAM) with $\boldsymbol{\theta}$ as its parameters. $w \in \mathcal{W} := \{w(\cdot, \cdot) : \mathcal{X} \times \mathcal{Y} \to [0, 1]\}$ is a re-weighting function that assign a weight to each sample. The bold $\boldsymbol{w}$ is a sample weight vector of length $N$ indicating training samples' importance, which contains per-sample weight $w_i := w(\boldsymbol{x}_i, y_i)$.

**Overall Framework.** Without the presence of clean validation set, our proposed bi-level optimization framework **SunGen** (as shown in Figure 2) can be outlined as follows:

$$\boldsymbol{w}^* \in \arg\min_{\boldsymbol{w}} \mathcal{L}_{\text{robust}}(\hat{\boldsymbol{\theta}}(\boldsymbol{w}), \mathcal{S}_{\text{syn}}^{\text{v}}) = \arg\min_{\boldsymbol{w}} \frac{1}{M} \sum_{\{\boldsymbol{x}, y\} \in \mathcal{S}_{\text{syn}}^{\text{v}}} \ell_{\text{robust}}(f(\boldsymbol{x}, \hat{\boldsymbol{\theta}}(\boldsymbol{w})), y) \quad (1)$$

$$s.t. \ \hat{\boldsymbol{\theta}}(\boldsymbol{w}) \in \arg\min_{\boldsymbol{\theta}} \mathcal{L}_{\text{ce}}(\boldsymbol{\theta}, \mathcal{S}_{\text{syn}}^{\text{t}}(\boldsymbol{w})) = \arg\min_{\boldsymbol{\theta}} \frac{1}{N} \sum_{\{\boldsymbol{x}, y\} \in \mathcal{S}_{\text{syn}}^{\text{t}}} w(\boldsymbol{x}, y) \ell_{\text{ce}}(f(\boldsymbol{x}, \boldsymbol{\theta}), y) \quad (2)$$

where $\boldsymbol{w}^*$ denotes the optimal sample weights, which is obtained from the outer loop (Eqn. 1); $\hat{\boldsymbol{\theta}}(\boldsymbol{w})$ denotes the classifier's parameters after weighted training with $\boldsymbol{w}$, which is obtained from the inner loop (Eqn. 2); $\ell_{\text{robust}}$ denotes the noise-robust loss calculated on validation set; $\ell_{\text{ce}}$ denotes the cross-entropy(CE) loss calculated on training set. The whole process is: (1) **in the inner loop**(Eqn. 2), we fix $\boldsymbol{w}$ and optimize the classifier $f(\boldsymbol{x}, \boldsymbol{\theta})$ using the weighted loss $\mathcal{L}_{\text{ce}}$ over a synthetic training set $\mathcal{S}_{\text{syn}}^{\text{t}}$, and derive the trained classifier $\hat{\boldsymbol{\theta}}(\boldsymbol{w})$; (2) **in the outer loop**(Eqn. 1), we calculate noise-robust loss $\mathcal{L}_{\text{robust}}$ on the synthetic validation set $\mathcal{S}_{\text{syn}}^{\text{v}}$ at the optimized $\hat{\boldsymbol{\theta}}(\boldsymbol{w})$. The outer loss is minimized to guide the optimization of $\boldsymbol{w}$, and the outer gradient $\nabla_{\boldsymbol{w}} \mathcal{L}_{\text{robust}}$ is calculated via truncated back-propagation(Appendix H). The two procedures are performed alternatively for $T$ iterations.

Notably, our framework prevents the need for a clean validation set, which is the main obstacle for the previous bi-level re-weighting approaches under zero-shot scenario (Ren et al., 2018; Shu et al., 2019). The magic of this appealing feature lies in the choice of the outer objective $\mathcal{L}_{\text{robust}}$ in Eqn. (1). Recall that the goal of sample reweighting is to find $\boldsymbol{w}^*$, with which the model $f(\boldsymbol{x}; \boldsymbol{\theta})$ can be trained using weighted loss over the synthetic training set, such that the model performs well on the clean data (from the same distribution as test data).

In Sec. 4, under the condition that the majority of data are correctly labelled (Ghosh et al., 2017; Wang et al., 2019), we theoretically show that using $\mathcal{L}_{\text{robust}}$ as the outer objective, our method can find a set of sample weights $\boldsymbol{w}^*$ with just the synthetic validation set, such that $\boldsymbol{w}^*$ maximizes the model performance on the clean data. The whole procedure is illustrated in Algorithm 1.

**Noise-robust Loss Functions.** In the noisy label learning literature, there is a family of loss functions that possess the following property:

$$\sum_{j=1}^K \ell_{\text{robust}}(f(\boldsymbol{x}, \boldsymbol{\theta}), j) = C, \forall \boldsymbol{\theta}, \boldsymbol{x}, \quad (3)$$

where $f(\cdot, \boldsymbol{\theta})$ denotes a classifier, $\boldsymbol{x}$ is the input, $K$ is the number of classes, and $C$ is a constant. Previous work (Wang et al., 2019; Zhang & Sabuncu, 2018; Ghosh et al., 2017) has shown that these loss functions have consistent global minimum under label noise. More formally, when the majority of the training samples are correctly

---

**Algorithm 1** Bilevel Robust Sample Reweighting

**Require:** a TAM with parameters $\boldsymbol{\theta}$, synthetic training dataset $\mathcal{S}_{\text{syn}}^{\text{t}}$ and validation dataset $\mathcal{S}_{\text{syn}}^{\text{v}}$, outer step size $\lambda$, outer iterations T.

1: Initialize sample weightings $\boldsymbol{w}$ with each $w_i$ to be 0.5.
2: **for** training iteration $t = 1, 2 \ldots T$ **do**
3:   Conduct weighted training ($\mathcal{L}_{\text{ce}}$) on TAM using $\boldsymbol{w}^t, \mathcal{S}_{\text{syn}}^{\text{t}}$ and obtain $\hat{\boldsymbol{\theta}}(\boldsymbol{w}^t)$
4:   Evaluate $\hat{\boldsymbol{\theta}}(\boldsymbol{w}^t)$ on $\mathcal{S}_{\text{syn}}^{\text{v}}$, then obtain meta gradient $\nabla_{\boldsymbol{w}} \mathcal{L}_{\text{robust}}$ via Eqn. (11).
5:   Update $\boldsymbol{w}$ using $\nabla_{\boldsymbol{w}} \mathcal{L}_{\text{robust}}$ by gradient descent $\boldsymbol{w}^{t+1} \leftarrow \boldsymbol{w}^t - \lambda \nabla_{\boldsymbol{w}} \mathcal{L}_{\text{robust}}$
6: **end forOutput:** Optimized sample weights $\boldsymbol{w}^*$.

---

labelled, the global minimizer ($\boldsymbol{\theta}^*$) of $\ell_{\text{robust}}$ is the same regardless of whether the training data is clean or noisy. This noise-robust property enables these losses to optimize the sample weights given only a noisy validation dataset. Detailed proof will be given in Sec. 4.

In particular, we consider the reversed cross-entropy loss $\ell_{\text{rce}}$ with the following form for sample $(\boldsymbol{x}, y)$:

$$\ell_{\text{rce}}(f(\boldsymbol{x}, \boldsymbol{\theta}), y) = -\sum_{k=1}^{K} f_k(\boldsymbol{x}, \boldsymbol{\theta}) \log q(k|\boldsymbol{x}),$$

where $f_k(\boldsymbol{x}, \boldsymbol{\theta}) = \frac{e^{z_k}}{\sum_{i=1}^{K} e^{z_i}}$ denotes the predicted probability for each label $k \in \{1, ..., K\}$, and $z_i$ are the logits. We denote the ground-truth distribution over labels by $q(k|\boldsymbol{x})$, and $\sum_{k=1}^{K} q(k|\boldsymbol{x}) = 1$. Given the ground-truth label is $y$, then $q(y|\boldsymbol{x}) = 1$ and $q(k|\boldsymbol{x}) = 0$ for all $k \neq y$. $log(0)$ is approximated to a constant $A$. One can easily check that $\ell_{\text{rce}}$ satisfies Property (3), and $C = -(K-1)A$ in this case.

**Remark.** Even though $\ell_{\text{robust}}$ is theoretically noise-tolerant, it has been shown that using them to train the network parameters $\boldsymbol{\theta}$ leads to difficulty in optimization and hampers the network's performance (Wang et al., 2019; Zhang & Sabuncu, 2018). Remarkably, adopting $\ell_{\text{robust}}$ as the objective in the outer loop of Eqn. (1) implicitly overcomes the side effects of $\ell_{\text{robust}}$. Since $\ell_{\text{robust}}$ is now used to optimize the sample weights $\boldsymbol{w}$, which is in a much smaller search space and has simpler structure than $\boldsymbol{\theta}$, thus is easier to optimize. This "decouples" noise removal from network training, and thereby prevents hampering the TAM's performance.

**Clean Subset Sampling.** Our framework enables us to derive a set of continuous weights $\boldsymbol{w}$, which encodes the data quality and can well separate the noisy samples from clean ones, as shown in Figure 3. With those weights, we can sample subsets with arbitrary budgets and use them to train the task models with unweighted CE loss. More specifically, suppose the budget is $D$, we first normalize $w_i$ to $w_i'$ (i.e. $\sum_{i=1}^{n} w_i' = D$), based on which we then sample a Bernoulli random variable $I_i \sim \mathbf{Ber}(w_i')$ for each sample to indicate whether they should be included. Denote the size of sampled subset as $\hat{D}$. Because $\mathbb{E}_{\boldsymbol{I} \sim p(\boldsymbol{I}|\boldsymbol{w})} \|\boldsymbol{I}\|_0 = \sum_{i=1}^{n} w_i' = D$, it is clear that $\frac{\hat{D}}{D} \xrightarrow{P} 1$ when $n \to \infty$, meaning that $\hat{D}$ is close to $D$. Notably, because the important and noise-free data are associated with larger weights, training on the sampled subset can perform on par with weighted training over the entire dataset, while being much more efficient.

## 4 THEORETICAL ANALYSIS

Though we have no clean data as validation set, our method still enjoys favorable theoretical properties. Recall that $\mathcal{S}_{\text{clean}}$ and $\mathcal{S}_{\text{syn}}$ are as the clean and synthetic (noisy) distribution, respectively. We further denote $\mathcal{L}(\boldsymbol{\theta}, \mathcal{S}) = \mathbb{E}_{(\boldsymbol{x},y) \sim \mathcal{S}}[\ell(f(\boldsymbol{x}; \boldsymbol{\theta}), y)]$ given a data distribution $\mathcal{S}$. Let the optimal network parameters obtained with CE loss over clean dataset be $\boldsymbol{\theta}^* := \arg\min_\theta \mathcal{L}_{\text{ce}}(\boldsymbol{\theta}, \mathcal{S}_{\text{clean}})$. We assume the $\boldsymbol{\theta}^*$ is unique, or we focus on the $\boldsymbol{\theta}^*$ with minimum norm.

**Property 1.** *We call a loss function $\ell$ a robust loss, if under mild conditions as in (Ghosh et al., 2017), its minimizer on the noisy dataset collides with the one on the clean dataset, i.e.,*

$$\arg\min_{\boldsymbol{\theta}} \mathcal{L}_{robust}(\boldsymbol{\theta}, \mathcal{S}_{clean}) = \arg\min_{\boldsymbol{\theta}} \mathcal{L}_{robust}(\boldsymbol{\theta}, \mathcal{S}_{syn}),$$

Previous work on noise-robust loss functions (Ghosh et al., 2017; Wang et al., 2019; Zhang & Sabuncu, 2018; Xu et al., 2019) have shown that the losses satisfying Eqn. (3) have the above noise-tolerant property. We give the detailed assumptions and complete proof in Appendix A.2.

**Assumption 1.** *Let $\mathbb{P}_{clean}$ and $\mathbb{P}_{syn}$ denote the probability density function of $\mathcal{S}_{clean}$ and $\mathcal{S}_{syn}$, respectively. There exists a weighting function $w^*$ such that $\mathbb{P}_{clean}(\boldsymbol{x}, y) = w^*(\boldsymbol{x}, y)\mathbb{P}_{syn}(\boldsymbol{x}, y)$.*

Assumption 1 is reasonable because synthetic data generated by the PLM has a wide coverage. Therefore, with a proper weight function $w$, we may recover the clean distribution by reweighting the synthetic data.

**Assumption 2.** *The optimal $\boldsymbol{\theta}^*$ uniquely minimizes $\mathcal{L}_{robust}(\boldsymbol{\theta}, \mathcal{S}_{clean})$, i.e., $\mathcal{L}_{robust}(\boldsymbol{\theta}^*, \mathcal{S}_{clean}) < \mathcal{L}_{robust}(\boldsymbol{\theta}, \mathcal{S}_{clean})$ for all $\boldsymbol{\theta} \neq \boldsymbol{\theta}^*$.*

Assumption 2 is natural since the robust losses are originally designed to train the model. Thus minimizing $\mathcal{L}_{\text{robust}}(\boldsymbol{\theta}, \mathcal{S}_{\text{clean}})$ is expected to achieve promising performance, as justified by Ghosh et al. (2017) (though the optimization difficulty mentioned in the Remark of Section 3 poses challenges for model training). We also experimentally show that when training a model with $\mathcal{L}_{\text{ce}}(\boldsymbol{\theta}, \mathcal{S}_{\text{clean}}^t)$, $\mathcal{L}_{\text{robust}}(\boldsymbol{\theta}, \mathcal{S}_{\text{clean}}^t)$ also decreases, and reaches the plateau at a similar point (Appendix A.1). This indicates that $\mathcal{L}_{\text{ce}}(\boldsymbol{\theta}, \mathcal{S}_{\text{clean}})$ and $\mathcal{L}_{\text{robust}}(\boldsymbol{\theta}, \mathcal{S}_{\text{clean}})$ have close optimal solutions. Note that even if $\boldsymbol{\theta}$ and $\boldsymbol{\theta}^*$ differ by a small quantity, i.e., $\mathcal{L}_{\text{robust}}(\boldsymbol{\theta}^*, \mathcal{S}_{\text{clean}}) < \mathcal{L}_{\text{robust}}(\boldsymbol{\theta}, \mathcal{S}_{\text{clean}}) + \varepsilon$ holds with a small $\varepsilon$, the following proof holds with just minor modifications.

**Theorem 1.** *If Assumption 1 holds, there exists a $\boldsymbol{w}^*$ such that $\hat{\boldsymbol{\theta}}(\boldsymbol{w}^*) = \boldsymbol{\theta}^*$. Further with Assumption 2 and Property 1, our method can uniquely return $\boldsymbol{w}^*$ and the resulting $\boldsymbol{\theta}^*$ with only synthetic (noisy) data $\mathcal{S}_{noisy}$.*

Given Assumption 2 holds, $\mathcal{L}_{\text{ce}}(\boldsymbol{\theta}, \mathcal{S}_{\text{clean}})$ and $\mathcal{L}_{\text{robust}}(\boldsymbol{\theta}, \mathcal{S}_{\text{clean}})$ have consistent optimal solution $\boldsymbol{\theta}^*$. Furthermore, Property 1 of $\mathcal{L}_{\text{robust}}$ indicates that $\boldsymbol{\theta}^*$ can be found with just the noisy synthetic data $\mathcal{S}_{\text{syn}}$. Therefore, we can optimize $\mathcal{L}_{\text{robust}}(\boldsymbol{\theta}, \mathcal{S}_{\text{syn}})$ to find $\boldsymbol{\theta}^*$. Finally, since $\boldsymbol{\theta}^*$ can be parameterized by $\boldsymbol{w}^*$ (Theorem 1), which is achieved by the inner loop (Eqn. (2)), we can optimize $\mathcal{L}_{\text{robust}}(\hat{\boldsymbol{\theta}}(\boldsymbol{w}), \mathcal{S}_{\text{syn}})$ over $\boldsymbol{w}$ in the outer loop (Eqn. (1)) to find the optimal corresponding $\boldsymbol{w}^*$ as well as the $\boldsymbol{\theta}^*$. See Appendix A.3 for detailed proof.

The above analysis theoretically shows our SUNGEN is able to learn the optimal sample weights $\boldsymbol{w}^*$ and the resulting optimal model parameters $\boldsymbol{\theta}^*$ with just synthetic data. Notably, Theorem 1 is based on population losses with infinite samples. We further characterize the generalization bound when there is finite samples:

**Theorem 2** (Finite Sample Generalization). *Suppose we have access to synthetic datasets $\mathcal{S}_{syn}^v$ and $\mathcal{S}_{syn}^t$ both with finite samples $N$. Let $\hat{\boldsymbol{\theta}}^*(\boldsymbol{w})$ be the deterministic mapping from $\boldsymbol{w}$ to $\boldsymbol{\theta}$ defined in the inner of Eqn.2 given $\hat{\mathcal{S}}_{syn}^t$. Assuming the output of loss function $\ell_{robust}$ is upper bounded by $M$, the carnality of $\mathcal{W}$ is $|\mathcal{W}|$, the outer loop of Eqn.1 is solved within $\epsilon$-approximately, and the solution $\hat{\boldsymbol{w}}$ satisfies the following condition:*

$$\mathcal{L}_{robust}(\hat{\boldsymbol{\theta}}^*(\hat{\boldsymbol{w}}); \mathcal{S}_{syn}^v) \leq \min_{\boldsymbol{w}} \mathcal{L}_{robust}(\hat{\boldsymbol{\theta}}^*(\boldsymbol{w}); \mathcal{S}_{syn}^v) + \epsilon,$$

*we then have with probability at least $1 - \delta$,*

$$\mathcal{L}_{robust}(\hat{\boldsymbol{\theta}}^*(\hat{\boldsymbol{w}}), \mathcal{S}_{syn}) \leq \min_{\boldsymbol{w}} \mathcal{L}_{robust}(\hat{\boldsymbol{\theta}}^*(\boldsymbol{w}); \mathcal{S}_{syn}) + \epsilon + \kappa \sqrt{\frac{2 \ln(|\mathcal{W}|/\delta)}{M}} \tag{4}$$

Refer to Appendix A.4 for full proof. Theorem 2 characterizes the generalization ability of SUNGEN. If we obtain an approximate solution of the bilevel problem Eqn.(1)-(2) given finite samples, Eqn. (4) shows that such solution has a test performance close to the oracle model.

## 5 EXPERIMENTS

### 5.1 SETUP

**Datasets & Baselines.** We evaluate SUNGEN on eight text classification tasks, including IMDb (Maas et al., 2011), SST-2 (Socher et al., 2013), Rotten Tomatoes (Pang & Lee, 2005), Amazon (McAuley & Leskovec, 2013), Yelp (Zhang et al., 2015), Subj (Pang & Lee, 2004), AGNews (Zhang et al., 2015) and DBpedia (Zhang et al., 2015). These tasks have various number of classes, ranging from 2 to 14. Other details about datasets are in Appendix F. We compare our proposed method with the following baselines: (1) **PROMPTING**. The prompt-based zero-shot classification method based on PLMs (Brown et al., 2020; Gao et al., 2021b). (2) **ZEROGEN**. A recent zero-shot learning work via dataset generation (Ye et al., 2022a).

**Implementation Details.** We compare the baselines using GPT2-XL (Radford et al., 2019) as PLM. For text generation, we use Nucleus Sampling (Holtzman et al., 2020) with $p = 0.9$ as the decoding

Table 1: Evaluation results for SUNGEN framework on two different scales of TAM. The scale of synthetic dataset is 200k for both ZEROGEN and SUNGEN. The scales of labeled data in supervised setting are listed under task names. *"Gold Data"* refers to the standard dataset with human annotations.

| TAM | #Param | Setting | IMDb | SST-2 | Rotten | Amazon | Yelp | Subj | AGNews | DBpedia | Avg |
|---|---|---|---|---|---|---|---|---|---|---|---|
| ***#Gold Data*** | | | *25k* | *6.9k* | *8.3k* | *25k* | *560k* | *8k* | *120k* | *560k* | *-* |
| DistilBERT | 66M | SUPERVISED | 87.24 | 89.68 | 83.67 | 92.63 | 95.42 | 95.95 | 94.51 | 99.14 | 92.28 |
| LSTM | ∼7M | | 84.60 | 76.30 | 77.49 | 86.36 | 91.30 | 90.20 | 90.61 | 98.28 | 86.89 |
| - | 1.5B | PROMPTING | 80.64 | 89.22 | 81.89 | 83.63 | 82.72 | 68.00 | 68.81 | 67.88 | 77.85 |
| DistilBERT | 66M | ZEROGEN | 84.28 | 87.27 | 83.02 | 87.19 | 87.58 | 80.45 | 76.48 | 79.32 | 83.20 |
| | | SUNGEN | **89.38** | **89.45** | **84.52** | **89.01** | **89.19** | **83.25** | **80.49** | **82.67** | **86.00** |
| LSTM | ∼7M | ZEROGEN | 71.52 | 74.89 | 73.45 | 80.48 | 84.95 | 69.15 | 73.34 | 67.02 | 74.35 |
| | | SUNGEN | **84.10** | **84.58** | **83.21** | **84.22** | **89.06** | **75.85** | **79.75** | **72.07** | **81.61** |

Table 2: Evaluation results using different validation sets ($\mathcal{S}^v$) in the outer loop. LSTM is used as TAM.

| Method | $\mathcal{S}^v$ | IMDb | Amazon | Yelp | Rotten |
|---|---|---|---|---|---|
| SUPERVISED | - | 84.60 | 86.36 | 91.30 | 77.49 |
| ZEROGEN | - | 71.52 | 80.48 | 84.95 | 73.45 |
| SUNGEN | Gold | 82.34 | **84.71** | 88.83 | 80.05 |
| | Syn. | **84.10** | 84.22 | **89.06** | **83.21** |

Table 3: Experimental comparison with other de-noise methods using LSTM as TAM.

| Method | IMDb | Amazon | Yelp |
|---|---|---|---|
| Confidence | 79.97 | 70.44 | 76.91 |
| Co-Teaching | 72.64 | 73.53 | 75.50 |
| Meta-Weight-Net | 71.23 | 79.25 | 83.41 |
| SUNGEN | **84.10** | **84.22** | **89.06** |

strategy and use GPT2-XL as the generator. To make a fair comparison, we use the best prompts designed by (Ye et al., 2022a) in both PROMPTING and data-generation settings (Appendix Table 10 ). For task model training, we use 1-layer Bi-LSTM and DistilBERT-base as the lightweight classifiers. The bilevel procedure is iterated 50 times for each task. For more details(e.g., full prompts, training details), please refer to Appendix F.

## 5.2 EXPERIMENTAL RESULTS

**Main Experiments.** We present our main experiment results in Table 1. We observe that our SUNGEN achieves considerable performance gain over the ZEROGEN baseline across all the tasks. Interestingly, the improvement is more prominent for LSTM compared with DistilBERT, which shows relative improvement over ZEROGEN by 9.8% on average across all tasks. We conjecture the reason is that the pre-trained models such as DistilBERT are inherently more robust to noisy training data, which is also pointed out in (Hendrycks et al., 2019). Surprisingly, on Rotten Tomatoes and Yelp, SUNGEN-LSTM even outperforms ZEROGEN-DistilBERT, which requires no pre-training and has much fewer parameters.

**Effectiveness of SUNGEN on Constructing Noise-Free Data.** Figure 1 shows that while ZeroGen suffers from overfitting on noisy data, the problem disappears when training with the subset selected by SUNGEN. With longer training epochs, the data of SUNGEN consistently helps improve the performance of TAM on test set and significantly surpasses the result of ZeroGen, which proves that SUNGEN can effectively construct noise-free high-quality data.

**Synthetic Data vs. Gold Data Supervision.** A key advantage of our proposed framework is that we do not require gold data to optimize sample weights. Here, we compare our SUNGEN to the bilevel reweighting method which uses gold data for calculating the outer objective in Table 2. Specifically, our SUNGEN calculates the robust outer objective $\ell_{rce}$ on the noisy synthetic data, while the counterpart calculates the normal outer objective $\ell_{ce}$ on gold data. Table 2 shows that our method achieves similar performances as the counterpart, which verifies that our method using the noise validation set can equivalently supervise the optimization of sample weights as the clean validation set does.

**SUNGEN vs. Popular Denoise Baselines.** We compare with other de-noise methods, which are (1) directly removing data with low predicted confidence (Swayamdipta et al., 2020), (2) Co-Teaching (Han et al., 2018), which distinguishes noisy data by large loss value, and (3) Meta-Weight-Net (Shu et al., 2019), which relies on meta samples and a loss-based proxy model to learn the sample weights. Since in zero-shot setting we have no access to gold data, we use the synthetic data as validation set for Co-Teaching and Meta-Weight-Net. Results in Table 3 prove our method's

Table 4: Results of SUNGEN-LSTM on different data sizes. Given subsets of SUNGEN, models are trained with $\ell_{ce}$. For the 1,000k set of SUN-GEN, the model is trained using weighted $\ell_{ce}$. Subsets that surpass the original full set (1,000k) are marked by **bold**. Performance of original full set is marked by underline.

| Size | IMDb | | Amazon | | Yelp | |
|---|---|---|---|---|---|---|
| | ZERO | SUN | ZERO | SUN | ZERO | SUN |
| 1,000k | 78.29 | 86.56 | 82.47 | 84.63 | 86.28 | 90.38 |
| 10k | 62.40 | 72.05 | 74.46 | 75.84 | 75.22 | 80.67 |
| 20k | 65.12 | **79.96** | 75.78 | 77.52 | 79.55 | 82.88 |
| 50k | 68.12 | **81.14** | 78.14 | 81.97 | 80.81 | 85.82 |
| 100k | 71.28 | **82.09** | 80.25 | **83.68** | 82.97 | **88.41** |
| 200k | 71.52 | **84.10** | 80.48 | **84.22** | 84.95 | **89.06** |

Table 5: Diversity and Correctness. We measure the *diversity* by Self-BLEU4 and *correctness* by accuracy of an oracle model (finetuned RoBERTa-Large). Lower Self-BLEU4 score indicates higher diversity. "SUNGEN-Top" and "SUNGEN-Bottom" represent 10k samples with highest and lowest weights respectively.

| Method | IMDb | Amazon | Yelp |
|---|---|---|---|
| *Diversity* ↓ | | | |
| Gold | 0.30 | 0.29 | 0.29 |
| ZEROGEN | 0.15 | 0.12 | 0.14 |
| SUNGEN | **0.14** | **0.10** | **0.11** |
| *Correctness(%)* ↑ | | | |
| Gold | 96.22 | 96.60 | 98.35 |
| ZEROGEN | 75.86 | 93.58 | 94.47 |
| SUNGEN | 82.27 | 88.87 | 90.78 |
| SUNGEN-Top | 86.00 | 84.20 | 85.33 |
| SUNGEN-Bottom | 4.57 | 61.50 | 44.66 |

superiority in zero-shot setting. We further analyze why the loss-value-based methods are not able to distinguish noisy data in our situation in Appendix L.

## 5.3 ABLATION STUDY AND ANALYSIS

**Performance of Different Data Sizes.** Table 4 shows that even with much less data, model trained by SUNGEN achieves better performance than ZEROGEN. For example, in IMDb, by removing low-quality data, subset with 2% data achieves better performance than the original data (20k vs. 1,000k), which shows superior data quality of SUNGEN. Besides, from Figure 3(d), we find the percentage of erroneous data is small. However, those erroneous data significantly degrades the model performance: without weighted training to remove the effect of noisy data, the IMDB accuracy of full set is decreased from 86.56 to 78.29.

**Analysis of Data Diversity and Correctness.** Table 5 shows that our selected noise-free dataset is more diverse than data generated by ZEROGEN. One interesting thing to note is that in Amazon and Yelp, average correctness of SUNGEN is slightly lower than ZEROGEN. In addition, the top samples (with highest weights) have slightly lower correctness than average. This is expected as the data with highest correctness may be redundant or too simple, while the challenging and informative samples are often harder to classify and thus have lower correctness. The results further verify that SUNGEN effectively separates the informative samples from the ones that are redundant or erroneous. This can not be done with the heuristic methods that manually sets a threshold to separate clean and noisy data, which may keep redundant samples and remove hard ones.

**Analysis of Removed and Selected Examples.** We take IMDb (sentiment classification task of movie review) as the example task and find that the majority of samples associated with small weights are wrongly labeled, as shown in Table 6. Besides, there is a small part of erroneous data which contains task-irrelevant text, meaning the generated text is not a movie review and has no obvious emotional tendency. For samples with large weights, we actually find them to be well-written transitional complex sentences (bottom part of Table 6), which verifies that SUNGEN tends to select correct and important samples.

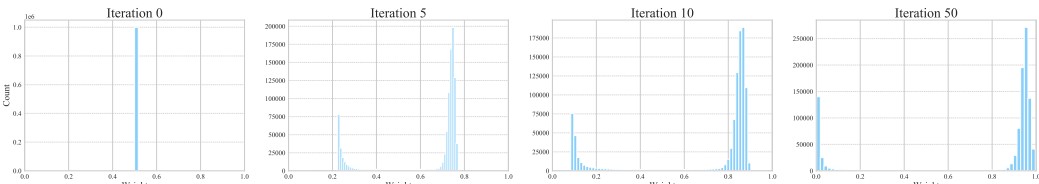

Figure 3: Histogram of learnt weights in IMDb synthetic dataset (1,000k). The weights are gradually separated as optimization proceeds, indicating SUNGEN can differentiate high-quality data from erroneous ones.

Table 6: Examples of removed data and selected data in IMDb synthetic dataset.

| Text<X> | Label<Y> | Noisy Type |
| --- | --- | --- |
| *Removed Data (Data with Small Weights)* | | |
| The film does a great job of capturing the fear and battle that so many U.S. troops have experienced during the seven-year war in Afghanistan. | Neg. | Noisy Y |
| This long, pompous, chain-smoking movie makes a big hit out of not very much at all. | Pos. | Noisy Y |
| One of the worst cult films ever made. 2D CGI animations of zombies and comic-book characters. Some bad acting, technical problems, cheap gimmicks and script that is. | Pos. | Noisy Y |
| The four-hour look at family dysfunction brings forth a richer character study. | Neg | Unrelated X |
| A fresh and visceral portrait of a man trying to make sense of his life. | Neg | Unrelated X |
| *Selected Data (Data with Large Weights)* | | |
| Despite its oddball structure and topsy-turvy interactions between characters, this surprisingly zany animated film succeeds where so many animated films fail. | Pos. | No Noise |
| Not worth the time for the main actors, but for the almost movie has a very good story that puts many sci-fi movies of the past to shame. | Pos. | No Noise |
| An outrageously stupid movie that has been thoroughly disproved by fact and logic. | Neg. | No Noise |
| Wonder Woman is a big-budget superhero blockbuster that turns the spotlight on the potential of a woman leader…but the movie is ultimately unfulfilling and laden with female stereotypes. | Neg. | No Noise |
| While a satire on class and power, the film's dismissal of human misery is shallow and the actors' portrayal of the weak and needy are gratingly self-pitying. | Neg. | No Noise |

## 6  RELATED WORK

**Zero-shot Learning via PLM.**    The popular prompt-based zero-shot prediction is proposed by GPT (Radford et al., 2019). With well-designed prompts, large-scale PLMs have shown its notable zero-shot learning ability in various tasks (Jiang et al., 2020; Shin et al., 2020; Reynolds & McDonell, 2021). More recently, data-generation-based work via PLM has gained popularity and shown superior ability on synthesizing task-specific data  (Anaby-Tavor et al., 2020; Puri et al., 2020; Kumar et al., 2020; Lee et al., 2021; Wang et al., 2021b; Yoo et al., 2021; Bonifacio et al., 2022). Apart from work that still relies on task-related human-annotated data to instruct or fine-tune the generative PLM, a recent line of research explores this direction in zero-shot scenario: Schick & Schütze (2021); Meng et al. (2022) use PLM with task-dependent prompts to generate data, and finetune another PLM on such data for task prediction. To further investigate PLM's zero-shot ability and alleviate the computation cost of PLM, Ye et al. (2022a) study an extreme scenario, which trains a tiny model from scratch using the synthetic data.

**Noise Robust Learning.**    Previous methods of tackling data noise can be categorized into two groups: (1) heuristic approaches based on loss values that rely on the assumption that the network learns easy samples first, which adopt either resampling (Han et al., 2018; Jiang et al., 2018; Yu et al., 2019), loss reweighting (Thulasidasan et al., 2019; Konstantinov & Lampert, 2019; Ren et al., 2018; Shu et al., 2019), or label correction (Ma et al., 2018; Kremer et al., 2018; Reed et al., 2014). These methods require either manually set a threshold for the loss value or a clean validation set, which makes their performance questionable in zero-shot scenario. (2) methods in another line train the network with noise-robust loss (Ghosh et al., 2017; Ma et al., 2020; Liu & Guo, 2020; Xu et al., 2019; Wang et al., 2019). Despite they learn a robust classifier in theory, they are typically difficult to train the DNNs and result require more hyper-parameter tuning (Zhang & Sabuncu, 2018; Wang et al., 2019). To this end, we take advantages from both lines of research and design an end-to-end framework which can reliably filter out harmful data without requiring a clean validation set.

## 7  CONCLUSION

This paper focuses on high-quality data generation in efficient zero-shot learning. To address the noise in data, we design an end-to-end framework to construct a clean synthetic dataset without relying on any human-labeled data or human intervention. Our method can be jointly applied to other data generation pipelines to automatically select high-quality data. We hope this paper can provide insights for improving the quality of synthetic datasets, and inspire more exploration in data-generation-based zero-shot learning via PLM.

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

## APPENDIX

## A PROOFS

### A.1 EMPIRICAL VERIFICATION FOR ASSUMPTION 2 AND 3

**Summary and Discussion.** In the sections below, we first show that (1) the poor performance of training the neural network with robust losses is due to the optimization difficulty, rather than the lack of good solution. We verify this by showing that the loss value of RCE is able to steadily decrease when the network is trained with CE loss. However, when training the network with RCE, the loss value fails to decrease. Then, we show that (2) the optimal solutions of CE and RCE losses are close by plotting the loss surface for both losses around the optimal solution obtained with CE loss. We are able to observe that the RCE loss is also close to the minimum around the optimal solution of CE loss. We believe the above two experiments can verify that the Assumption 2 and 3 are reasonable.

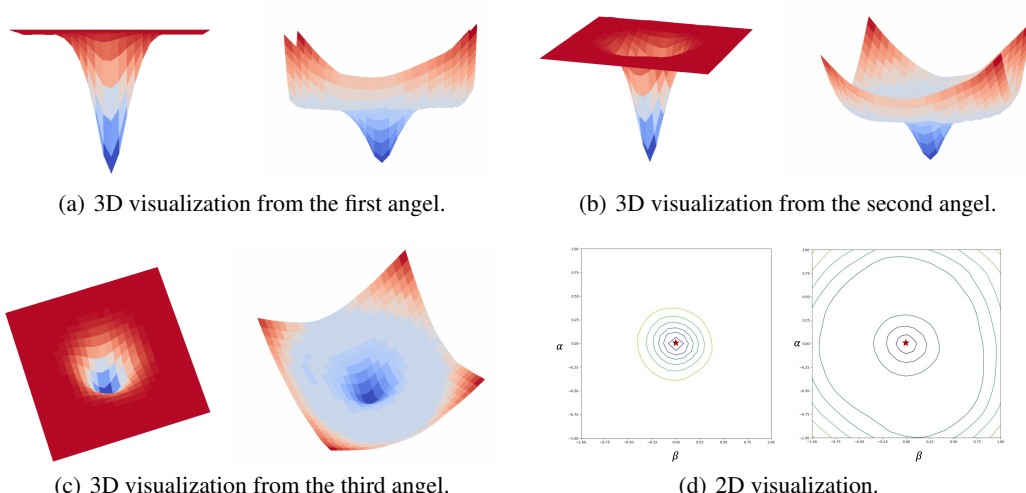

(a) 3D visualization from the first angel.
(b) 3D visualization from the second angel.

(c) 3D visualization from the third angel.
(d) 2D visualization.

Figure 4: Loss surface visualization of RCE and CE losses. We parameterize the surface with $\alpha$, $\beta$ parameters to vary model parameters and calculate its loss values alongside a 2D grid. In each subplot, the left one visualizes the loss surface of RCE loss; and the right one visualizes the CE loss surface. (a)-(c) are the visualization from different angles; (d) is the loss contour of CE and RCE.

### A.1.1  Optimization Difficulty of Noise-Robust Loss.

First, we analyze the cause of difficulty in the optimization of robust loss: (1) (Zhang & Sabuncu, 2018) show that the gradients of the cross-entropy loss have an implicit weighting term(as shown in Equation 5), which prioritizes the harder samples during training. On the other hand, this weighting does not exist in noise robust loss functions, which means these functions treat all samples equally. The lack of implicit weighting is claimed by (Zhang & Sabuncu, 2018) to be the cause of difficulty in training.

$$\sum_{i=1}^{n} \frac{\partial \mathcal{L}(f(\boldsymbol{x}_i; \boldsymbol{\theta}), y_i)}{\partial \boldsymbol{\theta}} = \begin{cases} \sum_{i=1}^{n} -\frac{1}{f_{y_i}(\boldsymbol{x}_i; \boldsymbol{\theta})} \nabla_{\boldsymbol{\theta}} f_{y_i}(\boldsymbol{x}_i; \boldsymbol{\theta}) & \text{for CE} \\ \sum_{i=1}^{n} -\nabla_{\boldsymbol{\theta}} f_{y_i}(\boldsymbol{x}_i; \boldsymbol{\theta}) & \text{for MAE/RCE.} \end{cases} \tag{5}$$

(2) Our experiment shows that the surface of the robust loss has a wide flat region, so when the parameters are not close to the optimal solution, the gradients could vanish, which leads to difficulty in optimization. The 3D visualization of the loss surface is shown in Figure 4. More details about drawing the figure are shown in A.1.2.

Second, we verify the performance degradation when training the network with robust loss is caused by optimization difficulty, rather than the lack of good solution. Without loss of generality, we compare the robust-loss $\ell_{\text{rce}}$ to $\ell_{\text{ce}}$. More specifically, we train the model with one loss, and use another loss to evaluate the model trajectory. From the results in Figure 5(a), we can clearly see that when optimizing the network with CE, both CE and RCE continue to decline throughout the training, and reach the plateau at the same time, which means that the RCE loss does indeed have a good solution. On the other hand, from the results in Figure 5(b), when training with RCE, both CE, and RCE hardly decrease, this further verifies that RCE is difficult to optimize when training the network.

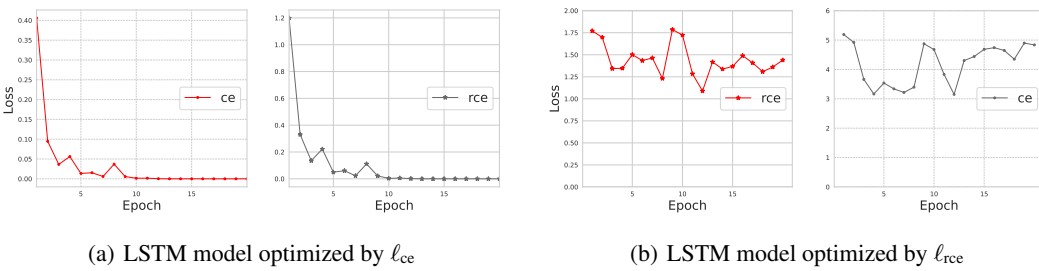

(a) LSTM model optimized by $\ell_{\text{ce}}$      (b) LSTM model optimized by $\ell_{\text{rce}}$

Figure 5: Loss curves of model training. Experiments run on the IMDb standard training set. The loss used to train the network is in red, and the other loss used to evaluate the network is in grey. We have run experiments using various learning rates from {1e-2, 1e-3, 1e-4, 1e-5} and the curves show similar trends.

### A.1.2  The Optimal Solutions of CE and RCE are Close.

Despite the optimization difficulty of RCE loss, the loss surfaces of CE and RCE losses indicate that the two loss functions have close optimal solutions. More specifically, we plot the loss surfaces of CE and RCE losses centered around the solution of CE loss following (Li et al., 2018), which visualizes the loss surfaces by modifying the model weights in two random directions:

$$f(\alpha, \beta) = L(\boldsymbol{\theta}^* + \alpha \boldsymbol{\delta} + \beta \boldsymbol{\eta})$$

where $\boldsymbol{\theta}^*$ is the model parameters optimized by CE, $\boldsymbol{\delta}$ and $\boldsymbol{\eta}$ are the random directions in the vector space of model parameters. $\alpha$ and $\beta$ are the scaling coefficients of the two directions, which control how far the parameters are perturbed. We parameterize the surface w.r.t the values of $\alpha$, $\beta$ ranging from [-1, 1], and calculate the loss values at different positions alongside a 2D grid. We show the visualization in Figure 4.

From sub-figure 4 (d), we can see that the optimal solution of CE is also close to the optimal solution of RCE (also has the lowest value near zero), which is direct experimental support for the Assumption 3. In addition, when the solution is not close to the optimal position, the rce loss surface is flat and wide, which is the cause of optimization difficulty.

## A.2 PROOF FOR PROPERTY 1

The noise-robust property of robust loss functions have been proven in previous work (Ghosh et al., 2017; Wang et al., 2019), we add them here for completeness. Specifically, we consider three cases of label noise: uniform noise, simple non-uniform noise and class-dependent noise following (Ghosh et al., 2017).

**Uniform Noise.** Given a classification problem with $K$ classes, for a loss function $\ell_{\text{robust}}$ satisfying Eqn.( 3). Then $\ell_{\text{robust}}$ is noise-tolerant under uniform label noise if the noise rate $\eta < \frac{K-1}{K}$. Let $\tilde{y}$ denote the noisy label from $\mathcal{S}_{\text{syn}}$, the proof is as follows:

$$
\begin{aligned}
\mathcal{L}_{\text{robust}}(\boldsymbol{\theta}, \mathcal{S}_{\text{syn}}) =& \mathbb{E}_{\boldsymbol{x}, \tilde{y}}[\ell_{\text{robust}}(fs(\boldsymbol{x}; \boldsymbol{\theta}), \tilde{y})] \\
=& \mathbb{E}_{\boldsymbol{x}, y} \mathbb{E}_{\tilde{y}|y, \boldsymbol{x}}[\ell_{\text{robust}}(f(\boldsymbol{x}; \boldsymbol{\theta}), \tilde{y})] \\
=& \mathbb{E}_{\boldsymbol{x}, y}[(1 - \eta)\ell_{\text{robust}}(f(\boldsymbol{x}; \boldsymbol{\theta}), y) + \frac{\eta}{K-1} \sum_{j \neq y}^{K} \ell_{\text{robust}}(f(\boldsymbol{x}; \boldsymbol{\theta}), j)] \\
=& \mathbb{E}_{\boldsymbol{x}, y}[\frac{K - 1 - K\eta}{K - 1} \ell_{\text{robust}}(f(\boldsymbol{x}; \boldsymbol{\theta}), y)] + \frac{\eta C}{K - 1} \\
=& \frac{K - 1 - K\eta}{K - 1} \mathcal{L}(\boldsymbol{\theta}, \mathcal{S}_{\text{clean}}) + \frac{\eta C}{K - 1}
\end{aligned}
$$

where $C$ is a constant due to the property of symmetric loss functions. Suppose $\boldsymbol{\theta}^*$ is the optimal solution for the clean dataset $\mathcal{S}_{\text{clean}}$. Therefore, we have the following inequality for any $\boldsymbol{\theta}$: $\mathcal{L}(\boldsymbol{\theta}^*, \mathcal{S}_{\text{syn}}) - \mathcal{L}(\boldsymbol{\theta}, \mathcal{S}_{\text{syn}}) = \frac{K-1-K\eta}{K-1}(\mathcal{L}(\boldsymbol{\theta}^*, \mathcal{S}_{\text{clean}}) - \mathcal{L}(\boldsymbol{\theta}, \mathcal{S}_{\text{clean}})) \leq 0$. Therefore, $\boldsymbol{\theta}^*$ is also the optimal solution for $\mathcal{S}_{\text{syn}}$.

**Class Dependent Noise.** For a loss function $\ell_{\text{robust}}$ satisfying Eq 3, and $0 \leq \ell_{\text{robust}}(f(\boldsymbol{x}; \boldsymbol{\theta}), i) \leq \frac{C}{K-1}, \forall i \in [K]$, and suppose $\min_{\boldsymbol{\theta}} \mathcal{L}(\boldsymbol{\theta}, \mathcal{S}_{\text{clean}}) = 0$. Then $\ell_{\text{robust}}$ is noise-tolerant under classs dependent label noise if $\eta_{ij} < 1 - \eta_i, \forall j \neq i, \forall i, j \in [K]$, $\eta_{ij}$ represents the probability of class $i$ mislabelled into class $j$. The proof is as follows:

$$
\begin{aligned}
\mathcal{L}_{\text{robust}}(\boldsymbol{\theta}, \mathcal{S}_{\text{syn}}) =& \mathbb{E}_{\boldsymbol{x}, \tilde{y}}[\ell_{\text{robust}}(f(\boldsymbol{x}; \boldsymbol{\theta}), \tilde{y})] \\
=& \mathbb{E}_{\boldsymbol{x}, y} \mathbb{E}_{\tilde{y}|y, \boldsymbol{x}}[\ell_{\text{robust}}(f(\boldsymbol{x}; \boldsymbol{\theta}), \tilde{y})] \\
=& \mathbb{E}_{\boldsymbol{x}, y}[(1 - \eta_y)\ell_{\text{robust}}(f(\boldsymbol{x}; \boldsymbol{\theta}), y) + \sum_{j \neq y}^{K} \eta_{yj} \ell_{\text{robust}}(f(\boldsymbol{x}; \boldsymbol{\theta}), j)] \\
=& \mathbb{E}_{\boldsymbol{x}, y}[(1 - \eta_y)(C - \sum_{j \neq y}^{K} \ell_{\text{robust}}(f(\boldsymbol{x}; \boldsymbol{\theta}), j)) + \sum_{j \neq y}^{K} \eta_{yj} \ell_{\text{robust}}(f(\boldsymbol{x}; \boldsymbol{\theta}), j)] \\
=& C\mathbb{E}_{\boldsymbol{x}, y}(1 - \eta_y) - \mathbb{E}_{\boldsymbol{x}, y} \sum_{j \neq y}^{K} (1 - \eta_y - \eta_{yj}) \ell_{\text{robust}}(f(\boldsymbol{x}; \boldsymbol{\theta}), j)
\end{aligned}
$$

Denote $\boldsymbol{\theta}^*$ and $\hat{\boldsymbol{\theta}}^*$ as $\arg\min_{\boldsymbol{\theta}} \mathcal{L}_{\text{robust}}(\boldsymbol{\theta}, \mathcal{S}_{\text{clean}})$ and $\arg\min_{\boldsymbol{\theta}} \mathcal{L}_{\text{robust}}(\boldsymbol{\theta}, \mathcal{S}_{\text{syn}})$, respectively. Given the above result, we have the following inequality for $\hat{\boldsymbol{\theta}}^*$ and $\boldsymbol{\theta}^*$:

$$
\mathcal{L}_{\text{robust}}(\hat{\boldsymbol{\theta}}^*, \mathcal{S}_{\text{syn}}) - \mathcal{L}_{\text{robust}}(\boldsymbol{\theta}^*, \mathcal{S}_{\text{syn}}) = \mathbb{E}_{\boldsymbol{x}, y}[\sum_{j \neq y}^{K} (1 - \eta_y - \eta_{yj})(\ell_{\text{robust}}(f(\boldsymbol{x}; \boldsymbol{\theta}^*), j) - \ell_{\text{robust}}(f(\boldsymbol{x}; \hat{\boldsymbol{\theta}}^*), j))] \leq 0
$$

Since $\ell_{\text{robust}}$ is always non-negative and $\mathcal{L}_{\text{robust}}(\boldsymbol{\theta}^*, \mathcal{S}_{\text{clean}}) = 0$, we have that $\ell_{\text{robust}}(f(\boldsymbol{x}; \boldsymbol{\theta}^*), y) = 0, \forall \boldsymbol{x}$. Thus, we have that $\ell_{\text{robust}}(f(\boldsymbol{x}; \boldsymbol{\theta}^*), i) = \frac{C}{K-1}, \forall i \neq y$ by the symmetric property of $\ell_{\text{robust}}$. Given the assumption on the label noise, $(1 - \eta_y - \eta_{yj}) > 0$. Therefore, for the equality to hold, we musts have $\ell_{\text{robust}}(f(\boldsymbol{x}; \hat{\boldsymbol{\theta}}^*, i) = \frac{C}{K-1}, \forall i \neq y$. According to the symmetric property of $\ell_{\text{robust}}$, we know that $\ell_{\text{robust}}(f(\boldsymbol{x}; \hat{\boldsymbol{\theta}}^*), y) = 0, \forall \boldsymbol{x}$. Which means that $\hat{\boldsymbol{\theta}}^*$ achieve zero losss on all clean samples as well. Therefore, $\hat{\boldsymbol{\theta}}^*$ is also the minimizer for $\mathcal{L}_{\text{robust}}(\boldsymbol{\theta}, \mathcal{S}_{\text{clean}})$.

**Non-Uniform Noise.** For a loss function $\ell_{\text{robust}}$ satisfying Eq 3, and suppose $\min_{\boldsymbol{\theta}} \mathcal{L}(\boldsymbol{\theta}, \mathcal{S}_{\text{clean}}) = 0$. Then $\ell_{\text{robust}}$ is noise-tolerant under non-uniform label noise if $\eta_x < \frac{K-1}{K}, \forall x$. The proof is as follows:

$$
\begin{aligned}
\mathcal{L}_{\text{robust}}(\boldsymbol{\theta}, \mathcal{S}_{\text{syn}}) =& \mathbb{E}_{\boldsymbol{x}, \tilde{y}}[\ell_{\text{robust}}(f(\boldsymbol{x}; \boldsymbol{\theta}), \tilde{y})] \\
=& \mathbb{E}_{\boldsymbol{x}, y} \mathbb{E}_{\tilde{y}|y, \boldsymbol{x}}[\ell_{\text{robust}}(f(\boldsymbol{x}; \boldsymbol{\theta}), \tilde{y})] \\
=& \mathbb{E}_{\boldsymbol{x}, y}[(1 - \eta_x)\ell_{\text{robust}}(f(\boldsymbol{x}; \boldsymbol{\theta}), y) + \sum_{j \neq y}^{K} \frac{\eta_x}{K-1} \ell_{\text{robust}}(f(\boldsymbol{x}; \boldsymbol{\theta}), j)] \\
=& \mathbb{E}_{\boldsymbol{x}, y}(1 - \eta_x)\ell_{\text{robust}}(f(\boldsymbol{x}; \boldsymbol{\theta}), y) + \mathbb{E}_{\boldsymbol{x}, y} \frac{\eta_x}{K-1}(C - \ell_{\text{robust}}(f(\boldsymbol{x}; \boldsymbol{\theta}), y)) \\
=& \mathbb{E}_{\boldsymbol{x}, y} \frac{C}{K-1} + \mathbb{E}_{\boldsymbol{x}, y}[(1 - \frac{K\eta_x}{K-1})\ell_{\text{robust}}(f(\boldsymbol{x}; \boldsymbol{\theta}), y)]
\end{aligned}
$$

Therefore, we have the following inequality for any $\boldsymbol{\theta}$: $\mathcal{L}_{\text{robust}}(\boldsymbol{\theta}^*, \mathcal{S}_{\text{syn}}) - \mathcal{L}_{\text{robust}}(\boldsymbol{\theta}, \mathcal{S}_{\text{syn}}) = \mathbb{E}_{\boldsymbol{x}, y}[(1 - \frac{K\eta_x}{K-1})(\ell_{\text{robust}}(f(\boldsymbol{x}; \boldsymbol{\theta}^*), y) - \ell_{\text{robust}}(f(\boldsymbol{x}; \boldsymbol{\theta}), y))]$. Since $\ell_{\text{robust}}$ is always non-negative and $\mathcal{L}_{\text{robust}}(\boldsymbol{\theta}^*, \mathcal{S}_{\text{clean}}) = 0$, we have that $\ell_{\text{robust}}(f(\boldsymbol{x}; \boldsymbol{\theta}^*), y) = 0, \forall \boldsymbol{x}$. Therefore, $\boldsymbol{\theta}^*$ is also the optimal solution for $\mathcal{S}_{\text{syn}}$.

### A.3  PROOF FOR THEOREM 1

*Proof.* We first show $\boldsymbol{\theta}^*$ exists in the space induced by $\hat{\boldsymbol{\theta}}(\boldsymbol{w})$. Then we show our framework uniquely return $\boldsymbol{\theta}^*$.

Step 1: Existence. By Assumption 1, we know that $\mathbb{P}_{syn}(\boldsymbol{x}, y)w^*(\boldsymbol{x}, y) = \mathbb{P}_{clean}(\boldsymbol{x}, y)$ in distribution. So

$$
\begin{aligned}
\hat{\boldsymbol{\theta}}(w^*) =& \arg\min_{\boldsymbol{\theta}} \mathcal{L}_{\text{robust}}(\boldsymbol{\theta}, \mathcal{S}_{\text{syn}}(w^*)) \\
=& \arg\min_{\boldsymbol{\theta}} \int \int w^*(\boldsymbol{x}, y)\ell((f(\theta, \boldsymbol{x}), y)\mathbb{P}_{syn}(\boldsymbol{x}, y)d\boldsymbol{x}d\boldsymbol{y} \\
=& \arg\min_{\boldsymbol{\theta}} \mathbb{E}_{\mathbb{P}_{syn}(\boldsymbol{x}, y)w^*(\boldsymbol{x}, y)}\ell((f(\theta, \boldsymbol{x}), y) \\
=& \arg\min_{\boldsymbol{\theta}} \mathbb{E}_{\mathbb{P}_{clean}(\boldsymbol{x}, y)}\ell(f(\boldsymbol{\theta}, \boldsymbol{x}), y) \\
=& \boldsymbol{\theta}^*.
\end{aligned}
$$

Step 2: Uniqueness. By Assumption 2, we have for all $\boldsymbol{\theta} \neq \boldsymbol{\theta}^*$, we have $\mathcal{L}_{\text{robust}}(\boldsymbol{\theta}^*, \mathcal{S}_{clean}) < \mathcal{L}_{\text{robust}}(\boldsymbol{\theta}, \mathcal{S}_{clean})$. So $\boldsymbol{\theta}^* = \arg\min_{\boldsymbol{\theta}} \mathcal{L}_{\text{robust}}(\boldsymbol{\theta}, \mathcal{S}_{clean})$. Since $\arg\min_{\boldsymbol{\theta}} \mathcal{L}_{\text{robust}}(\boldsymbol{\theta}, \mathcal{S}_{syn}) = \arg\min_{\boldsymbol{\theta}} \mathcal{L}_{\text{robust}}(\boldsymbol{\theta}, \mathcal{S}_{clean})$. So we have

$$
\boldsymbol{\theta}^* = \arg\min_{\boldsymbol{\theta}} \mathcal{L}_{\text{robust}}(\boldsymbol{\theta}, \mathcal{S}_{syn}).
$$

Putting the existence and uniqueness parts together, we finish the proof. $\qquad\square$

The above theorem shows that there is an explicit mapping between the optimal weighting function $w^*$ and the optimal $\boldsymbol{\theta}^*$. This mapping is achieved by the inner loop in formulation 2. We can then optimize $\mathcal{L}_{\text{ce}}(\boldsymbol{\theta}(\boldsymbol{w}), \mathcal{S}_{\text{clean}})$ over $\boldsymbol{w}$ to find the optimal $\boldsymbol{w}^*$. Given that Assumption 2 holds, we can switch $\mathcal{L}_{\text{ce}}(\boldsymbol{\theta}(\boldsymbol{w}), \mathcal{S}_{\text{clean}})$ to $\mathcal{L}_{\text{robust}}(\boldsymbol{\theta}(\boldsymbol{w}), \mathcal{S}_{\text{clean}})$. Finally, thanks to Property 1 of $\ell_{\text{robust}}$, we may simply optimize $\mathcal{L}_{\text{robust}}(\boldsymbol{\theta}(\boldsymbol{w}), \mathcal{S}_{\text{syn}})$ over $\boldsymbol{w}$ instead. This gives rise to the outer loop in formulation 1.

### A.4  PROOF FOR THEOREM 2

*Proof.* Let $\hat{\mathcal{S}}_{syn}^{v,-1}$ denotes the dataset that replaces any one element of $\hat{\mathcal{S}}_{syn}^{v}$ with arbitrary $\boldsymbol{x}$, it is easy to know that

$$
|\mathcal{L}_{\text{robust}}(\hat{\boldsymbol{\theta}}^*(\boldsymbol{w}); \mathcal{S}_{\text{syn}}^{v}) - \mathcal{L}_{\text{robust}}(\hat{\boldsymbol{\theta}}^*(\boldsymbol{w}); \mathcal{S}_{\text{syn}}^{v,-1})| \leq \frac{\kappa}{M}
$$

holds for any $\boldsymbol{w}$. Then by the bounded difference inequality (Corollary 2.21 of Wainwright (2019)), given $\boldsymbol{w}$, we have with probability $1 - \delta$,

$$\mathcal{L}_{\text{robust}}(\hat{\boldsymbol{\theta}}^*(\boldsymbol{w}); \mathcal{S}_{\text{syn}}) \leq \mathcal{L}_{\text{robust}}(\hat{\boldsymbol{\theta}}^*(\boldsymbol{w}); \mathcal{S}^v_{\text{syn}}) + \kappa\sqrt{\frac{\ln(1/\delta)}{2M}}, \tag{6}$$

Then we have

$$\mathcal{L}_{\text{robust}}(\hat{\boldsymbol{\theta}}^*(\hat{\boldsymbol{w}}); \mathcal{S}_{\text{syn}})$$

$$\leq \mathcal{L}_{\text{robust}}(\hat{\boldsymbol{\theta}}^*(\hat{\boldsymbol{w}}); \mathcal{S}^v_{\text{syn}}) + \kappa\sqrt{\frac{2\ln(|\mathcal{W}|/\delta)}{M}}$$

$$\leq \mathcal{L}_{\text{robust}}(\hat{\boldsymbol{\theta}}^*(\boldsymbol{w}); \mathcal{S}^v_{\text{syn}}) + \kappa\sqrt{\frac{\ln(|\mathcal{W}|/\delta)}{2M}} + \epsilon$$

$$\leq \mathcal{L}_{\text{robust}}(\hat{\boldsymbol{\theta}}^*(\boldsymbol{w}); \mathcal{S}^v_{\text{syn}}) + \kappa\sqrt{\frac{\ln(1/\delta)}{2M}}$$

$$+ \kappa\sqrt{\frac{\ln(|\mathcal{W}|/\delta)}{2M}} + \epsilon$$

$$\leq \mathcal{L}_{\text{robust}}(\hat{\boldsymbol{\theta}}^*(\boldsymbol{w}); \mathcal{S}_{\text{syn}}) + \kappa\sqrt{\frac{2\ln(|\mathcal{W}|/\delta)}{M}} + \epsilon,$$

The first inequality because we require inequality equation 6 to hold uniformly for all $|\mathcal{W}|$ functions. The second inequality is because $\hat{\boldsymbol{w}}$ is the $\epsilon$-approximated solution. The third inequality is applying inequality equation 6. The forth inequality is because $|\mathcal{W}| > 1$. Taking infimum over $\boldsymbol{w}$ on the right hand side, we obtain the desired bound. □

## B  RELAXING ASSUMPTION 2

Here we provide additional results on relaxed assumptions. We relax Assumption 2 as follows:

**Assumption 3.** *The optimal $\boldsymbol{\theta}^*$ achieves $\epsilon$-optimal robust loss $\mathcal{L}_{robust}(\boldsymbol{\theta}, \mathcal{S}_{clean})$ on the clean dataset, i.e., $\mathcal{L}_{robust}(\boldsymbol{\theta}^*, \mathcal{S}_{clean}) < \mathcal{L}_{robust}(\boldsymbol{\theta}, \mathcal{S}_{clean}) + \epsilon$ for all $\boldsymbol{\theta} \neq \boldsymbol{\theta}^*$ with $\epsilon > 0$.*

Then we have the following results that extend Theorem 1:

**Theorem 3.** *If Assumption 1 holds, there exists a $\boldsymbol{w}^*$ such that $\hat{\boldsymbol{\theta}}(\boldsymbol{w}^*) = \boldsymbol{\theta}^*$. Further with Assumption 3 and Property 1 and assume $\mathcal{L}_{robust}(\boldsymbol{\theta}(\boldsymbol{w}), \mathcal{S}_{clean})$ is $\mu$-strongly convex and differentiable w.r.t. $\boldsymbol{w}$, our method can return a $\boldsymbol{w}$ that is close to $\boldsymbol{w}^*$ as follows:*

$$\|\boldsymbol{w} - \boldsymbol{w}^*\|_2 \leq \sqrt{2\epsilon/\mu}.$$

*Further, if the minimizer of the robust loss on the clean data collides with $\boldsymbol{\theta}^*$, i.e., $\epsilon = 0$, we have $\boldsymbol{w} = \boldsymbol{w}^*$ and $\boldsymbol{\theta} = \boldsymbol{\theta}^*$.*

*Proof.* Similar to the first step of the proof of Theorem 1 in Appendix A.3, we know $\hat{\boldsymbol{\theta}}(\boldsymbol{w}^*) = \boldsymbol{\theta}^*$. Further by Property 1 we know the solution of $\arg\min_{\boldsymbol{\theta}} \mathcal{L}_{\text{robust}}(\boldsymbol{\theta}, \mathcal{S}_{\text{syn}})$ is the same with that of $\arg\min_{\boldsymbol{\theta}} \mathcal{L}_{\text{robust}}(\boldsymbol{\theta}, \mathcal{S}_{\text{clean}})$. We further have

$$\arg\min_{\boldsymbol{w}} \mathcal{L}_{\text{robust}}(\hat{\boldsymbol{\theta}}(\boldsymbol{w}), \mathcal{S}_{\text{syn}}) = \arg\min_{\boldsymbol{w}} \mathcal{L}_{\text{robust}}(\hat{\boldsymbol{\theta}}(\boldsymbol{w}), \mathcal{S}_{\text{clean}}).$$

Denote $\bar{\boldsymbol{w}} = \arg\min_{\boldsymbol{\theta}} \mathcal{L}_{\text{robust}}(\hat{\boldsymbol{\theta}}(\boldsymbol{w}), \mathcal{S}_{\text{clean}})$, it follows that

$$\nabla_{\boldsymbol{w}} \mathcal{L}_{\text{robust}}(\bar{\boldsymbol{\theta}}(\boldsymbol{w}), \mathcal{S}_{\text{clean}}) = 0 \tag{7}$$

By Assumption 3, we then have

$$\mathcal{L}_{\text{robust}}(\boldsymbol{\theta}^*, \mathcal{S}_{\text{clean}}) \leq \mathcal{L}_{\text{robust}}(\hat{\boldsymbol{\theta}}(\bar{\boldsymbol{w}}), \mathcal{S}_{\text{clean}}) + \epsilon. \tag{8}$$

We further have

$$\mathcal{L}_{\text{robust}}(\boldsymbol{\theta}^*, \mathcal{S}_{\text{clean}}) = \mathcal{L}_{\text{robust}}(\hat{\boldsymbol{\theta}}(\boldsymbol{w}^*), \mathcal{S}_{\text{clean}})$$

$$\geq \mathcal{L}_{\text{robust}}(\hat{\boldsymbol{\theta}}(\bar{\boldsymbol{w}}), \mathcal{S}_{\text{clean}}) + \nabla_{\boldsymbol{w}} \mathcal{L}_{\text{robust}}(\hat{\boldsymbol{\theta}}(\bar{\boldsymbol{w}}), \mathcal{S}_{\text{clean}})(\boldsymbol{w}^* - \bar{\boldsymbol{w}})$$

$$+ \frac{\mu}{2}\|\boldsymbol{w}^* - \bar{\boldsymbol{w}}\|_2^2 \tag{9}$$

The equality in the first line is due to $\hat{\boldsymbol{\theta}}(\boldsymbol{w}^*) = \boldsymbol{\theta}^*$. The inequality comes from the strong convexity assumption. Combining Eqn equation 7, equation 8 and equation 9, we have

$$\frac{\mu}{2}\|\boldsymbol{w}^* - \bar{\boldsymbol{w}}\|_2^2 \leq \epsilon.$$

The result follows immediately by noting that $\bar{\boldsymbol{w}}$ is the output of our method. □

## C ABLATION STUDY OF OUTER OBJECTIVES

In the main paper, we directly use RCE as an example of noise-robust loss to verify our framework. Here we conduct additional experiments using the other noise-robust Mean Absolute Error (MAE) loss (Ghosh et al., 2017). The results in Table 7 show that SUNGEN framework using RCE and MAE losses both achieve promising performance, which significantly surpasses baseline methods. Besides, if we consider the standard cross-entropy(CE) loss as the outer objective, the bi-level framework can only achieve marginal improvement or even worse than ZEROGEN. The result is reasonable as the standard CE loss in the outer loop run on synthetic data, which cannot provide accurate guidance to update the per-sample weights.

Table 7: Experimental comparison using different outer objectives. Since no clean validation set exists in the zero-shot setting, all the outer objectives are calculated on synthetic data. The experiments run on 200k synthetic data using LSTM as TAM.

| Method | Outer Objective | IMDb | Amazon | Yelp |
|---|---|---|---|---|
| ZEROGEN | - | 71.52 | 80.48 | 84.95 |
| Bi-level | CE | 74.05 | 79.81 | 83.90 |
| SUNGEN | RCE | 84.10 | 84.22 | 89.06 |
| | MAE | 84.23 | 84.05 | 89.23 |

## D SUNGEN VS. OTHER NOISE-ROBUST TRAINING STRATEGIES

We compare our method with the label smoothing and temporal ensemble noise-training strategies in Table 8. The experimental results show that our method achieved a significant improvement over these two counterparts. In addition, label smoothing and temporal ensemble are training strategies to alleviate the noise issue, which cannot be used to select a high-quality subset.

Table 8: Comparison between SUNGEN with other noise-robust training strategies. Experiments run on 200k synthetic data using LSTM as TAM.

| Method | IMDb | Amazon | Yelp |
|---|---|---|---|
| ZEROGEN | 71.52 | 80.48 | 84.95 |
| Label Smoothing | 73.18 | 81.91 | 86.07 |
| Temporal Ensemble | 74.10 | 81.42 | 85.82 |
| SUNGEN | 84.10 | 84.22 | 89.06 |

## E EXPERIMENTS ON OTHER CLASSIFICATION TASKS

As suggested, we conduct experiments on more difficult tasks of GLUE in Table 9, including the NLI tasks (RTE, QNLI) and paraphrase task (MRPC). The experimental results show that SunGen can consistently surpass the baseline methods. For more challenging tasks that need expert knowledge or reasoning ability, we leave them as our future work.

Table 9: Comparison between SUNGEN with other noise-robust training strategies. Experiments run on 20k synthetic data using DistillBERT as TAM.

| Method | RTE | QNLI | MRPC |
|---|---|---|---|
| PROMPTING | 54.51 | 60.60 | 65.04 |
| ZEROGEN | 57.04 | 65.46 | 68.71 |
| SUNGEN | 62.82 | 71.82 | 71.25 |

# F   IMPLEMENTATION DETAILS

## F.1   DATASET

We evaluate our method on eight widely-used text classification tasks, including IMDb (Maas et al., 2011), SST-2 Socher et al. (2013), Rotten Tomatoes (Pang & Lee, 2005), Amazon (McAuley & Leskovec, 2013), Yelp (Zhang et al., 2015), Subj (Pang & Lee, 2004), AGNews (Zhang et al., 2015) and DBpedia (Zhang et al., 2015). IMDB, SST-2, and Rotten Tomatoes are sentiment classification benchmarks containing positive/negative movie reviews. Amazon and Yelp are comments classification tasks consisting of electronic product reviews and restaurant reviews respectively. We choose electronics and restaurant reviews as they are very different from movie reviews. Subj is a subjectivity detection task to justify whether the text contains factual contents or expresses opinions. AGNews (4-class classification) and DBpedia (14-class classification) are the topic and otology classification tasks respectively. Apart from AGNews and DBpedia, other tasks are binary classification tasks. We use full test set for evaluation except for DBpedia, for which we randomly sample 5000 test examples to reduce computational cost. Sample sizes are listed in Table 1. We report accuracy for evaluation.

## F.2   FULL IMPLEMENTATION DETAILS

We compare the baselines using GPT2-XL (Radford et al., 2019) as PLM. For text generation, we use Nucleus Sampling (Holtzman et al., 2020) with $p = 0.9$ as the decoding strategy and use GPT2-XL as the generator. For fair comparison, we use the best prompts designed by (Ye et al., 2022a) for data generation. During the optimization of sample weights, we use Adam optimizer. For selecting the appropriate value of the outer learning rate, we select from {2.5e-1, 1e-1, 1e-2} by looking at the value of RCE loss in the outer loop. If the outer loss steadily decreases and reaches a low value, then it indicates that the optimization is going well. In the inner loop, 1,000k synthetic data are used as the training data; in the outer loop, 50k synthetic samples are randomly sampled as the training data for fast iteration. We use 1-layer Bi-LSTM and DistilBERT-base as the tiny task model and run it for 1 epoch each time for fast iteration. The bilevel procedure is iterated for 50 times for each task.

For task model training, we use 1-layer Bi-LSTM and DistilBERT-base as the light-weight classifiers. For LSTM, we use Adam optimizer(Kingma & Ba, 2015) with learning rate 1e-3. For DistilBERT-base, we finetune each dataset using Adam optimizer with learning rate 2e-5, and other default hyper-parameters as suggested by HuggingFace Transformers library(Wolf et al., 2019). We run LSTM for 5 epochs, and run DistilBERT-base for 3 epochs for prediction. Unless otherwise stated, we run our experiments on 200k data. We compute the average accuracy on test set over 3 runs using different random seeds.

For the baseline using gold data in Table 2, to simulate the scenario where gold data is scarce, we randomly select 1,000 samples from the standard training set as the training data in outer loop. For comparison with other denoising baselines shown in Table 3, we use the techniques as described in the original papers. Specifically, for Confidence (Swayamdipta et al., 2020), we use the mean model probability of the true label across epochs as the confidence value and select top 200k examples; for Co-teaching (Han et al., 2018), we use two networks, each is trained with samples selected by the other network based on the loss value; for meta-weight-net (Shu et al., 2019), since we do not have access to clean data, we use a part of synthetic data as validation. Co-teaching and meta-weight-net are conducted on 200k synthetic data.

### F.3 PROMPTS

For IMDb, SST-2, and Rotten Tomatoes, we use the best prompts designed by Ye et al. (2022a) in both PROMPTING and data-generation settings. For tasks that are not investigated by ZEROGEN, following Ye et al. (2022a), we manually design five prompts for each task and report the result of the best prompt on PROMPTING, then we use the same prompt(or minor revision version) for ZEROGEN and SUNGEN to generate data. The details of prompts are shown in Table 10.

Table 10: Prompts used for PROMPTING and data generation. Note that ZEROGEN and SUNGEN use the same prompt for each task. <c> represents the input movie names/electronic categories/restaurant names. *<MASK>* position will be placed with label words. <x> represents the text that we use PLM to generate. The movie and restaurant names are generated by PLM using prompt *"Movie: "*, *"Restaurant: "*. The electronic categories are 41 categories collected from Amazon website.

| Setting | Task | Best Prompt | Label Words |
|---|---|---|---|
| PROMPTING | IMDb | The movie review in *<MASK>* sentiment is: "<x> | positive/negative |
| | Amazon | The electronics product review in *<MASK>* sentiment is: "<x> | positive/negative |
| | Yelp | The restaurant review in *<MASK>* sentiment is: "<x> | positive/negative |
| | SST-2 | The movie review in *<MASK>* sentiment is: "<x> | positive/negative |
| | Rotten | The movie review in *<MASK>* sentiment is: "<x> | positive/negative |
| | Subj | The movie review *<MASK>* is: "<x> | containing factual contents (objective) / expressing opinions (subjective) |
| | AGNews | The news article is in the category of *<MASK>*: "<x> | World/Sports/Business/ Technology |
| | DBpedia | The article classified to the category of *<MASK>*: "<x> | Company/School/Artist/ Athlete/ Politician/ Transportation/ Building/ Nature/Village/Animal/Plant/ Album/ Film/Book |
| ZEROGEN & SUNGEN | IMDb | The movie review in *<MASK>* sentiment for movie "<c>" is: "<x> | positive/negative |
| | Amazon | The "<c>" product review in *<MASK>* sentiment is: "<x> | positive/negative |
| | Yelp | The "<c>" restaurant review in *<MASK>* sentiment is: "<x> | positive/negative |
| | SST-2 | The movie review in *<MASK>* sentiment for movie "<c>" is: "<x> | positive/negative |
| | Rotten | The movie review in *<MASK>* sentiment for movie "<c>" is: "<x> | positive/negative |
| | Subj | The movie review *<MASK>* is: "<x> | containing factual contents (objective) / expressing opinions (subjective) |
| | AGNews | The news article is in the category of *<MASK>*: "<x> | World/Sports/Business/ Technology |
| | DBpedia | The article classified to the category of *<MASK>*: "<x> | Company/School/Artist/ Athlete/ Politician/ Transportation/ Building/ Nature/Village/Animal/Plant/ Album/ Film/Book |

## G BI-LEVEL VS. ONE-LEVEL OPTIMIZATION

We empirically verify the effectiveness of bi-level SUNGEN than one-level optimization using $l_{\text{rce}}$ (*One-level, $\ell_{rce}$*). From the results in Table 11, we can observe that our framework with bi-level $\ell_{\text{rce}}$ outperforms both one-level counterparts significantly.

Table 11: Comparison between our method with one-level $\ell_{\text{rce}}$ on 1,000k data. LSTM is used as TAM.

| Method | IMDb | Amazon | Yelp |
|---|---|---|---|
| *One-level, $\ell_{rce}$* | 81.92 | 81.50 | 85.36 |
| SUNGEN | 86.56 | 84.63 | 90.38 |

## H  TRUNCATED BACK-PROPAGATION FOR META GRADIENT.

For solving the bilevel optimization problem, the gradient of $w$ can be calculated as follows:

$$
\begin{aligned}
&\nabla_{\boldsymbol{w}} \mathcal{L}_{\text{robust}} \\
&= \nabla_{\boldsymbol{\theta}} R_{\text{robust}}|_{\theta*} \ \nabla_{\boldsymbol{w}} \hat{\boldsymbol{\theta}}(\boldsymbol{w}) \\
&= \nabla_{\boldsymbol{\theta}} \mathcal{L}_{\text{robust}}|_{\theta_T} \sum_{j \leq T} \left[ \prod_{k<j} I - \frac{\partial^2 \mathcal{L}_{\text{ce}}}{\partial \boldsymbol{\theta} \partial \boldsymbol{\theta}^{\mathsf{T}}} \bigg|_{\boldsymbol{\theta}_{T-k-1}} \right] \frac{\partial^2 \mathcal{L}_{\text{ce}}}{\partial \boldsymbol{\theta} \partial \boldsymbol{w}^{\mathsf{T}}} \bigg|_{\boldsymbol{\theta}_{T-j-1}} \qquad (10) \\
&\approx \nabla_{\boldsymbol{\theta}} \mathcal{L}_{\text{robust}}|_{\theta_T} \ \frac{\partial^2 \mathcal{L}_{\text{ce}}}{\partial \boldsymbol{\theta} \partial \boldsymbol{w}^{\mathsf{T}}} \bigg|_{\boldsymbol{\theta}_{T-1}}, \qquad\qquad\qquad\qquad\qquad\qquad\qquad (11)
\end{aligned}
$$

where Eqn. (10) follows chain rule. For computational efficiency, we do not unroll the entire $T$ steps, but perform 1-step truncated backpropagation as in Eqn. (11) (Shaban et al., 2019).

## I  COMPARISON WITH OTHER NOISE-ROBUST LEARNING METHODS

Our framework has the following advantages against other noise-robust learning methods:

- Compared with heuristic methods (Han et al., 2018; Jiang et al., 2018), our framework is end-to-end and does not require excessive manual tuning and task-specific knowledge.

- Since both the inner and outer objectives are calculated on the same synthetic training set, we do not need any in-domain labeled data, which is a must in the previous end-to-end reweighting methods (Ren et al., 2018; Shu et al., 2019).

- Compared with methods that train the model with $\ell_{\text{robust}}$ (Zhang & Sabuncu, 2018; Wang et al., 2019), our approach leverages $\ell_{\text{robust}}$ to learn the sample weights, which enables removing the low-quality data without hurting the model's performance

## J  MORE RELATED WORK

**Bilevel Optimization**    Bilevel optimization (BO) Sinha et al. (2017), which is what we rely upon when building our algorithm, has received much attention recently. This optimization technique has the ability to tackle problems with hierarchical structures. BO has been successfully adopted in numerous applications, such as hyper-paramter optimization Lorraine et al. (2020); Maclaurin et al. (2015); MacKay et al. (2019); Franceschi et al. (2017); Vicol et al. (2021), neural architecture search Pham et al. (2018); Liu et al. (2018); Pham et al. (2018); Shi et al. (2020); Yao et al. (2021); Gao et al. (2022; 2021a); Shi et al. (2021), meta learning Finn et al. (2017); Nichol & Schulman (2018), dataset condensation Wang et al. (2018); Zhao et al. (2020); Cazenavette et al. (2022); Pi et al. (2022) and sample re-weighting Ren et al. (2018); Shu et al. (2019); Zhou et al. (2022a;b).

## K  SELECTED AND REMOVED DATA

The selected and removed examples are listed in Table 6. We take IMDb as the example task. The observation shows that most of the removed data (data with low weights) have noise label, which indicates the class of text is wrongly labeled by PLM during generation(Noisy Y). Besides, there is a small part of erroneous data which contains unrelated text to the task, meaning the generated text is not a movie review showing obvious emotional tendency(Unrelated X). From Figure 3(d), we find the percentage of the erroneous data is small, but it significantly degrades the model performance(e.g., IMDB accuracy is decreased from 86.56 to 78.29 in Table 4). For selected data by SUNGEN, we find they are actually well-written transitional complex sentences (bottom part of Table 6), which verifies that SUNGEN tends to select correct and challenging samples.

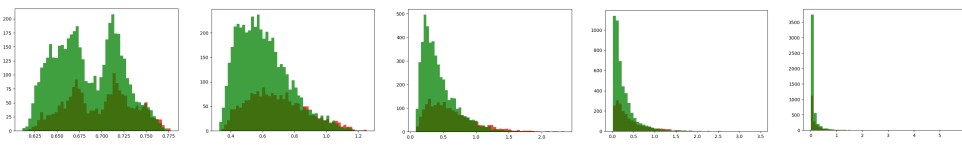

(a) Epoch 0(54%).    (b) Epoch 1( 61%).    (c) Epoch 2(63%).    (d) Epoch 3(60%).    (e) Epoch 4(62%).

Figure 6: Loss histogram of SST-2 with 30% uniform label noise. Clean data and noisy data are marked by green and red respectively. The accuracy is listed in the parentheses. We observe that the loss value can not separate the clean data from the noisy ones. Therefore, the methods loss-value-based methods may not work well in our case.

## L  WHY LOSS-VALUE-BASED DE-NOISE METHODS DOES NOT WORK IN OUR SITUATION?

We further empirically analyze why the popular loss-value-based methods (Shu et al., 2019; Han et al., 2018) fail to help sample noise-free dataset in our scenario. For this experiment, we manually construct a dataset with a 30% label noise using the training set of SST-2 by randomly flipping the class label. As shown in figure 6, the loss values of the correctly labelled data and the mislabelled data are still clustered together. We conjecture that this is due to the nature of the task: not all tasks demonstrate the phenomenon that noisy data demonstrate higher loss values, which is also mentioned in works related to instance dependent noise learning (Cheng et al., 2020). Therefore, selecting subset based on the loss value is not applicable in our scenario.

