# OpenReview forum: "Self-Guided Noise-Free Data Generation for Efficient Zero-Shot Learning"
_ICLR.cc/2023/Conference — ICLR 2023 notable top 25%_

### Official Review · Reviewer_23fs · 2022-10-22

**Confidence:** 4
**Correctness:** 4
**Technical Novelty And Significance:** 3
**Empirical Novelty And Significance:** 3
**Recommendation:** 8

**Clarity, Quality, Novelty And Reproducibility:**

* Clarity: The paper is overall clear and well organized.
* Quality: The contribution of the paper is interesting, targeting an important and timely topic in zero-shot learning. The empirical performance is also good. Nevertheless, there are a few missing empirical evaluations that could have made the contribution more convincing (see cons above). There are also concerns regarding the theoretical analyses that the authors may need to consider revising (see cons above).
* Novelty: The paper proposes a novel approach for generation-based zero-shot learning built on meta-learning without requiring manually labeled clean dev sets.

**Strength And Weaknesses:**

Pros:
* The paper targets an important and timely topic in zero-shot learning -- generation-based approaches recently have become a promising and popular direction for zero-shot learning, but the major challenge of how to effectively leverage the potentially noisy synthetic dataset has remain unresolved. The proposed method that applies the idea of meta-learning to automatically learn sample weights without requiring manually-labeled clean dev sets is new and interesting.
* The method is empirically effective across eight classification tasks, outperforming the ZeroGen baseline, and even achieving comparable performance to fully-supervised training that uses the clean training data (though they are mostly of smaller size than the generated dataset).
* The paper is overall clearly written and well organized.

Cons:
* The theoretical analyses are not informative. I believe that Assumption 2 is too strong (or even wrong) and makes the entire proofs trivial. Assumption 2 essentially states that the optimal solution on the clean set induced by the noise robust loss function is exactly the same with that induced by the standard cross-entropy loss. However, previous studies (e.g., Zhang and Sabuncu) have shown that noise robust losses usually worsen the converged solution compared to standard cross-entropy when used on clean datasets, so it's hardly the truth that they actually lead to similar optima. I believe Assumption 2 is actually what requires more formal proofs (Theorems 1 & 2 will trivially hold given Assumption 2). The empirical validation presented in D.1 is not convincing since similar training curve trends do not imply similar global/local optima.
* The choice of noise robust loss function needs to be further articulated. The paper directly picks reversed cross-entropy loss without explaining the reason. There are many functions that can satisfy Eq. (3) -- even a constant loss function that does nothing for training. It is therefore necessary to justify why reversed cross-entropy loss is used (rather than other noise robust loss functions in, e.g., Ghosh et al. & Zhang and Sabuncu). This may be done empirically as ablation studies.
* It would have made the empirical evaluation more complete if the method was compared with other noise-robust training strategies (e.g., label smoothing & temporal ensembling used in Meng et al.) and on more difficult classification tasks (e.g., the NLU tasks from the GLUE benchmark).
* (Minor) There are a few typos and unnatural language usages that need further proofreading. For example, (1) "despite we can generate unlimited training data in theory" -> "despite the unlimited training data that one can generate in theory" (2) "we may recover the clean distributional" -> "we may recover the clean distribution" (3) "the bi-level optimization approaches has been proven" -> "the bi-level optimization approaches have been proven" (4) "To attack the noise data" -> "To address the noise in data"
* (Minor) There seem to be too much whitespace at the bottom of each page -- the authors may need to follow the official template for formatting.

---
**Post-Rebuttal Updates**:
I'd like to thank the authors for providing very detailed responses and revisions to address my raised concerns above. I especially liked the additional analyses w.r.t. the noise-robust loss function, including its comparison to the standard cross entropy loss in terms of their optimization difficulty & optimal solutions, the empirical effectiveness of the chosen loss function (RCE loss) over other noise-robust training strategies and other alternative loss functions, and the relaxed theoretical assumption (Assumption 3 and Theorem 3). More detailed comments are below:
* It's interesting to see that the noise-robust loss functions result in different optimization difficulties compared to the standard cross-entropy loss, though it's still a bit unclear to me whether their optimal solutions would be actually the same (or at least very similar). The loss surface plots and training curves are definitely good empirical evidence, but I'd be interested to know a more principled answer to this (of course, this is obviously beyond the scope of this paper, and I won't count this as a weakness of the paper at all).
* Even if the optimal solutions induced by the noise-robust loss function concur with those of the standard cross-entropy loss, the **actual converged solutions** may differ due to optimization difficulties. Therefore, I believe that a more relaxed assumption (Assumption 3) is indeed necessary here, and I'm happy to see the new theoretical analyses built on this.
* The new results on the three GLUE tasks and the ablations for comparing different loss functions/noise-robust training strategies look good to me.
* The authors' responses to my further clarification questions all make sense to me. I think it'll be good to discuss the efficiency and time costs in the next paper version (I understand that the time cost is affordable due to the small classification model used, but it'll be beneficial to let the readers have a sense of how the method scales to larger models/datasets). Other than that, there is only one point that I don't quite get:
"Besides, our training process is “once for all”. In other words, once we derive the sample weights, we can use them to sample high-quality subsets at different scales."
-> Do you intend to say that you could apply some filtering method based on the learned sample weights to select a subset of the synthetic training set in order to improve training efficiency?

Based on the above considerations, I believe that the updated paper has addressed almost all of my concerns (the remaining issues are very minor and can be apparently fixed easily). Overall, this is a good paper with an important research goal, clear presentations, novel ideas, solid implementations, good theoretical insights, and promising empirical performance. I'll be happy to support the paper for acceptance.

**Summary Of The Paper:**

The paper studies data generation-based zero-shot learning that mainly uses a large PLM to generate training data and then trains a small tiny task model (TAM) to perform classification. The major challenge that the paper tries to address is that the synthetic data are usually noisy, and hence the paper proposes a meta-learning empowered framework to automatically learn sample weights over the synthetic dataset -- the goal is to upweight the accurate samples and downweight the noisy ones. Different from previous meta-learning approaches that assume a clean validation set for learning sample weights, this paper uses a noise-robust loss function calculated on the synthetic dataset for the outer-loop optimization, thus removing the requirement for manually annotated data. The proposed method SunGen is compared with ZeroGen on eight classification tasks and demonstrates significant improvements.

**Summary Of The Review:**

I appreciate the attempt to address the major challenge with leveraging synthetic datasets generated by PLMs -- the existence of label noise. The proposed method is also novel and interesting. However, there are a few points (e.g., strong theoretical assumptions, missing discussions of loss function choices, and other empirical evaluations) that need further revisions to be made convincing.

---

> ### Author Response · Authors · 2022-11-15
> **Rely to Reviewer 23fs (Part 1/3)**
>
> We sincerely thank Reviewer 23fs for the review and are grateful for the time you spent with our submission. We are glad for the acknowledgment that our approach is new and interesting to tackling an important and timely topic.   We wish to address your concerns by giving detailed responses to each of your comments as follows:
>
>
> **Q1 “Assumption 2 is too strong. (Zhang and Sabuncu) show that noise robust losses worsen the converged solution compared to CE on clean dataset. Assumption 2 requires more proof. ”**
>
> Thank you for your comments, we agree that more support evidence is needed for Assumption 2, and we provide them below. Besides, since the response block cannot show figures, **we have attached the related figures including important analysis(e.g. 3D loss surface visualization) to Appendix C** in the latest version.  For your convenience, **we also put the answers with plots at the [anonymous link](https://randomname2023.github.io/)**.
>
> 1. **The claim made by (Zhang and Sabuncu) is not that the robust loss functions lead to worse converged solutions, but they are actually harder to converge from the optimization point of view.** We support the point from two aspects:
> - **First, the cause of difficulty in the optimization of robust loss are**: **(1)** In the mentioned paper(Zhang and Sabuncu), they show that the gradients of the cross-entropy loss have an implicit weighting term(as shown in the following equation), which is inversely proportional to the ground-truth label’s corresponding entry at the classifier’s output. In other words, the CE loss prioritizes the harder samples during training. On the other hand, this weighting does not exist in noise robust loss functions, which means these functions treat all samples equally. The lack of implicit weighting is what (Zhang and Sabuncu) claim to be the cause of difficulty in training. **(2)** Our experiment shows that the surface of the robust loss has a wide flat region, so when the parameters are not close to the optimal solution, the gradients could vanish, which leads to difficulty in optimization (this is verified via more extensive experiments below). For the 3D visualization of the loss surface, please refer to Appendix Figure 4 in the current version. More details about drawing the figure are shown in Appendix  C.1.2.
> >$$\sum^n_{i = 1}\frac{\partial L(f(x_i ; \theta), y_i)}{\partial \theta} = \begin{cases} \sum^n_{i = 1} - \frac{1}{f_{y_i}(x_i ; \theta)}\nabla_{\theta} f_{y_i}(x_i ; \theta), \text{for CE;} \\\\\\sum^n_{i = 1} - \nabla_{\theta} f_{y_i}(x_i ; \theta), \text{for RCE/MAE.}
> \end{cases}$$
>
>
>
>
>
> - Second, we verify **the performance degradation when training the network with robust loss is caused by optimization difficulty, rather than the lack of good solution**, via the following experiment:  We train two networks on standard IMDb training set: one with CE loss and another with RCE loss, and for both networks, we track the values of CE and RCE losses calculated over the training set (please refer to Appendix Figure 5). We can clearly see that when optimizing the network with CE, both CE and RCE continue to decline throughout the training, and reach the plateau at the same time, which means that the RCE loss does have a good solution. On the other hand, when training with RCE, both CE, and RCE hardly decrease, this further verifies that RCE is difficult to optimize when training the network.
>
> 2. **Theoretically, we further relax Assumption 2 to Assumption 3, which only requires the optimal solution of CE loss to also achieve a reasonably small value for RCE loss**, which is within $\epsilon$ difference from the optimal value of RCE loss. Based on Assumption 3, we have extended Theorem 1 to Theorem 3 and added detailed proof in Appendix B. Assumption 3 is also supported by the following experiments.
>
> >**Assumption 3.** The optimal $\boldsymbol{\theta}^*$ achieves reasonable robust loss $L_{robust}(\boldsymbol{\theta}, S_{clean})$ on the clean daraset, i.e., there exists $\epsilon>0$ such that  $L_{robust}(\boldsymbol{\theta}^*, S_{clean})<L_{robust}(\theta, S_{clean})+\epsilon$ for all $\boldsymbol{\theta}^* \neq \boldsymbol{\theta}^*$ .
> >
> Then we have the following results that extend Theorem 1:
> > **Theorem 3.** If Assumption holds, there exists a $\boldsymbol{w}^*$ such that $\hat{\boldsymbol{\theta}}(\boldsymbol{w}^*) = \boldsymbol{\theta}^*.$
> > Further with Assumption 3 and Property 1 and assume $L_\text{robust}(\boldsymbol{\theta}(\boldsymbol{w}), S_\text{clean})$ is $\mu$-strongly convex and differentiable w.r.t. $\boldsymbol{w}$ , our method can return a $\boldsymbol{w}$ that is close to $\boldsymbol{w}^*$ as follows:
> >
> > $$\|\boldsymbol{w} - \boldsymbol{w}^*\|_2 \leq \sqrt{2\epsilon/\mu}$$
> >
> > Further, if the minimizer of the robust loss on the clean data collides with $\boldsymbol{\theta}^*$, i.e.,  $\epsilon=0$, \text{we have}
> > $\boldsymbol{w} = \boldsymbol{w}^*$ and $\boldsymbol{\theta}=\boldsymbol{\theta}^*$.

---

> > ### Author Response · Authors · 2022-11-15
> > **Rely to Reviewer 23fs (Part 2/3)**
> >
> > **Part 2/3**
> >
> > The Remaining Answer for Q1:
> >
> > 3. **The loss surfaces of CE and RCE losses show the two loss functions have close optimal solutions** (please refer to  Figure 4 in the Appendix C).  More specifically, we plot the loss surfaces of CE and RCE losses centered around the optimal solution of CE loss following  (Li et al., 2018),  which visualizes the loss surface by perturbing the network parameters along two randomly sampled directions. The procedure can be formulated as: $f(\alpha,\beta)=L(\boldsymbol \theta^*+\alpha \boldsymbol \delta+\beta \boldsymbol \eta) \nonumber$,
> > where $\theta^*$ is the model parameters trained to convergence with CE loss, $\delta$ and $\eta$ are the random directions in the vector space of model parameters. $\alpha$ and $\beta$ are the scaling coefficients of the two directions, which control how far the parameters are perturbed. We parametrize the surface w.r.t the values of α, β ranging from [-1, 1], and calculate the loss values at different positions alongside a 2D grid. **From the plot (Appendix Figure 4(d)), we can see that the optimal solution of CE is also close to the optimal solution of RCE** , which is direct experimental support for Assumption 3. In addition, when the solution is not close to the optimal position, the rce loss surface is flat and wide, which is the cause of optimization difficulty.
> >
> > We also want to note that apart from the theoretical analysis, our SunGen indeed brings considerable performance boosts over the competing methods(as shown in the main experiments). We believe our proposed framework is a good contribution to the PLM-based zero-shot learning direction.
> >
> > Many thanks for the good suggestion. We have attached the above analysis and the further relaxation of Assumption 2 in the current version appendix.
> >
> > **Q2 “The choice of noise robust loss function needs to be further articulated. The paper directly picks reversed cross-entropy loss without explaining the reason. There are many functions that can satisfy Eq. (3) -- even a constant loss function that does nothing for training. It is therefore necessary to justify why reversed cross-entropy loss is used (rather than other noise robust loss functions in, e.g., Ghosh et al. & Zhang and Sabuncu). ”**
> >
> > Thank you for pointing this out. In the last version, we use RCE as the example of noise-robust loss to verify our framework, since it has been thoroughly investigated in previous work and demonstrates promising performance. We agree that it is necessary to compare different noise-robust losses. We hope to address your concerns in the following:
> >
> > 1. We conduct extra experiments by replacing the RCE loss with Mean Absolute Error(MAE) loss (Ghosh et al. & Zhang and Sabuncu)  as the outer objective of SunGen. The experiment results show that the two losses demonstrate similar performances. For completeness, we have added the experiment to the paper. (Appendix A)
> >
> > 2. We want to note that we adopt RCE in our method because it has been well-established and analyzed in the noisy label learning field, which has a neat form that shares a similar flavor as CE loss and also demonstrates promising results.
> >
> > |   Method        |  IMDb |Amazon |Yelp
> > |  ----           | ----  | ----  | ----
> > |      ZeroGen    | 71.52 | 80.48 |84.95
> > |     SunGen-rce  | 84.10 |84.22  |89.06
> > |     SunGen-mae  | 84.23 |84.05  |89.23
> >
> >  *(Experiments run on 200k synthetic data. )*
> >
> > **Q3 “Compared with other noise-robust training strategies”**
> >
> > We add label smoothing and temporal ensemble noise-training strategies in the following table. The experimental results show that our method achieved a significant improvement over label smoothing and temporal ensembling training strategies. In addition, label smoothing and temporal ensembling are the training strategies, which have some limitations that cannot be used to select a high-quality subset. We have added the comparison in Appendix D.
> >
> > |   Method        |  IMDb |Amazon |Yelp
> > |  ----           | ----  | ----  |----
> > |      ZeroGen    |  71.52| 80.48|84.95
> > | label-smoothing |  73.18| 81.91|86.07
> > | temporal-ensemble |  74.10| 81.42|85.82
> > |      SunGen     | 84.10 |84.22 |89.06
> >
> >  *(Experiments run on 200k synthetic data. )*
> >
> > **Q4 “ More difficult classification tasks.”**
> >
> > As suggested, we conduct experiments on more difficult tasks in GLUE, including the NLI tasks (RTE, QNLI) and paraphrase task (MRPC). The experimental results show that SunGen can consistently surpass the baseline methods.  We agree with the reviewer that it is still a great challenge for tasks that need expert knowledge or reasoning ability. We think this is a promising future direction. We have added this part in Appendix E.
> >
> > |   Method          |  RTE  | QNLI  | MRPC
> > |  ----                  | ----  |  ---- | ----
> > | Prompting   | 54.51 | 60.60 |65.04
> > | ZeroGen | 57.04 | 65.46 |68.71
> > | SunGen  | 62.82 | 71.82 |71.25
> >  (Experiments run on 20k synthetic data.)

---

> > > ### Author Response · Authors · 2022-11-15
> > > **Rely to Reviewer 23fs (Part 3/3)**
> > >
> > > **Q5. Typo and format.**
> > >
> > > Thanks for kindly pointing out our typos and format problem. We have revised them all in the latest version.
> > >
> > > Overall, many thanks for your insightful points and suggestions. These comments really help improve our work and make the justification of our method more complete. We have added the experiments and revised the theoretical analysis accordingly in our current version. We hope our answers have addressed your concerns. If you have any further questions, we are happy to address them. We would really appreciate it if you are willing to increase your score.
> > >
> > >
> > > **Reference:**
> > >
> > > [1] Li, H., Xu, Z., Taylor, G., Studer, C., & Goldstein, T. (2018). Visualizing the loss landscape of neural nets. Advances in neural information processing systems, 31.
> > >
> > > [2] Ghosh, A., Kumar, H., & Sastry, P. S. (2017, February). Robust loss functions under label noise for deep neural networks. In Proceedings of the AAAI conference on artificial intelligence (Vol. 31, No. 1).
> > >
> > > [3] Zhang, Z., & Sabuncu, M. (2018). Generalized cross entropy loss for training deep neural networks with noisy labels. Advances in neural information processing systems, 31.

---

> > > > ### Author Response · Authors · 2022-11-17
> > > > **Further comments and discussions will be appreciated!**
> > > >
> > > > Dear Reviewer 23fs,
> > > >
> > > > Thank you for your valuable time to review our work and constructive feedback. We posted our response to your comments two days ago, and we wonder if you could kindly share some of your thoughts so we can keep the discussion rolling to address your concern if there are any.
> > > >
> > > > In the previous response,
> > > >
> > > > 1. We further relax Assumption 2 to Assumption 3 and theoretically extend Theorem 1 to Theorem 3, which only requires the optimal solution of CE loss to also achieve a reasonably small value for RCE loss.
> > > > 2. We conduct additional experiments to justify that Assumption 3 is reasonable by visualizing the loss surface of CE and RCE losses. The experiments are in the appendix of the current version.
> > > > 3. We conduct extra experiments by replacing the RCE loss with Mean Absolute Error(MAE) loss (Ghosh et al. & Zhang and Sabuncu)  as the outer objective of SunGen, showing that the two losses demonstrate similar performances.
> > > > 4. We add experiments of label smoothing and temporal ensemble noise-training strategies. The results show that SunGen achieved a significant improvement over other strategies.
> > > > 5. As suggested, we conduct experiments on more difficult tasks in GLUE, including the NLI tasks (RTE, QNLI) and paraphrase task (MRPC). The results show that SunGen can consistently surpass the baseline methods.
> > > >
> > > > We would appreciate it if you could kindly take a look at both the revision and our response to your comments. If you have any further questions, we are happy to discuss them!
> > > >
> > > > Best regards,
> > > >
> > > > Authors

---

> > > > ### Author Response · Authors · 2022-11-21
> > > > **Reply to Reviewer 23fs - Supplementary to Q1**
> > > >
> > > > Dear Reviewer 23fs,
> > > >
> > > > We would like to thank you again for your detailed and constructive reviews. To better answer your Q1 “Assumption 2 requires more proofs”,  we have **added a dynamic loss surface visualization** and supplemented analysis with corresponding plots. For your convenience, we further put the results with plots **at the [anonymous link](https://randomname2023.github.io/)**.
> > > >
> > > > For other questions, we have updated our draft and added replies to your comments with our latest experimental results. We would appreciate it if you could let us know if our responses have addressed your concerns and whether you still have any other questions about the current draft. We would be happy to do any follow-up discussion or address any additional comments.
> > > >
> > > > Thanks very much!
> > > >
> > > > Best regards,
> > > >
> > > > Authors

---

> ### Comment · Reviewer_23fs · 2022-11-21
> **Quick Update**
>
> I'd like to thank the authors for updating the paper and providing answers to my questions. This is intended to be a quick update to the authors that I'm aware of the responses -- please allow me some time to read them through. As I indicated in my original review, I quite liked the paper and I'll be happy to upgrade my rating if my concerns are addressed in the updated paper version.
>
> In the meantime, I'd like the authors to comment on the following (note that they are more of clarification questions than weaknesses/concern about the paper, and apologize if they are already addressed in the authors responses which I haven't got a chance to fully go through):
> * I'd like to see a brief summary of the differences between the two zero-shot data generating methods, Meng et al. & Ye et al. I understand that their central ideas are similar, but a brief discussion about their differences will help justify why Ye et al. is used as the baseline instead of Meng et al.
> * Meta weighting learning details: In Section 3, the authors mentioned that the sample weights are learned with both the data sample and its label as the feature (i.e., $w_i = w(\boldsymbol{x}_i, y_i)$); how exactly is this implemented? Do you use the bag-of-words feature for the input sequence $\boldsymbol{x}$ or use some distributed representations obtained from some encoder for it? And the input sequence feature will be concatenated with the corresponding categorical label as the input to the weighting network?
> * Time efficiency of model training & data generation process: Meta weight learning algorithms (e.g., Shu et al.) usually incur 3x training time of the standard training. Is this also the case for SunGen? Besides, it looks like one needs to generate quite a lot of synthetic training data for good performance. I'd be curious to know the time cost for the data generation process as well.
>
> Again, I just hope the authors could briefly comment on the above questions (I'm not asking for any extra thorough experiments). Thank you again for the updates and looking forward to the authors' further response.

---

> > ### Author Response · Authors · 2022-11-23
> > **Reply to Quick Update**
> >
> > Dear Reviewer 23fs,
> >
> > We sincerely appreciate your quick update and the kind words that you quite like our paper. Besides, many thanks for your additional constructive comments.  We are glad to have further discussion on your comments here, and we wish to address your concerns by giving detailed responses to each of your comments below:
> >
> > **Q1. I'd like to see a brief summary of the differences between the two zero-shot data generating methods, Meng et al. & Ye et al. I understand that their central ideas are similar, but a brief discussion about their differences will help justify why Ye et al. is used as the baseline instead of Meng et al.**
> >
> > Many thanks for pointing out this question. Meng et al. & Ye et al. are concurrent works that propose the generation-based zero-shot learning paradigm. Their data generation approaches are similar. The difference is that Ye et al. use the synthetic data to train a tiny classifier (e.g. LSTM with 6M parameters) to investigate the extreme scenario of efficient zero-shot learning, while Meng et al. train a large classifier (e.g. pre-trained COCO-LM-Large/RoBERTa-Large with 360M parameters) to explore the full potential of PLM’s zero-shot learning ability.  We choose the ZeroGen paradigm since it is more efficient and requires less computation cost. We will clarify the differences between these two works in the final version.
> >
> >
> >  **Q2. Meta weighting learning details: In Section 3, the authors mentioned that the sample weights are learned with both the data sample and its label as the feature (i.e., $w_i=w(x_i,y_i)$); how exactly is this implemented? Do you use the bag-of-words feature for the input sequence x or use some distributed representations obtained from some encoder for it? And the input sequence feature will be concatenated with the corresponding categorical label as the input to the weighting network?**
> >
> > Thanks for pointing out this potential confusion. We adopt the function notation for the convenience of theoretical analysis（Proof for Theorem 1).  In our implementation, the $w_i$ is actually implemented by real value instead of a function,  meaning that each sample has a separate weight and all weights are equally initialized to 0.5. Then our bi-level framework will directly optimize the vector $(w_1, w_2, …w_n)$ over the training process. The weights can finally measure the quality of $(x_i,y_i)$. Please refer to Algorithm 1 for more details. In our scenario, we optimize the weights after accumulating a large amount of generated data. Although the design is simple, we experimentally find direct optimizing $w_i$ demonstrates superior performance.
> >
> > However, optimizing a proxy function to generate the weights is a promising direction for future work, especially for some online scenarios. In such a situation, the proxy's structure and input feature should be more carefully designed.  We will clarify this confusion in the final version.
> >
> > **Q3. Time efficiency of model training & data generation process: Meta weight learning algorithms (e.g., Shu et al.) usually incur 3x training time of the standard training. Is this also the case for SunGen? Besides, it looks like one needs to generate quite a lot of synthetic training data for good performance. I'd be curious to know the time cost for the data generation process as well.**
> >
> > **Model Training.** Since we use a tiny task model as the classifier (e.g. LSTM), which is efficient for training and inference, the whole bi-level training time is relatively short. More specifically, we only need 2 GPU hours to finish the 50-iteration iterative updating. Besides, our training process is “once for all”. In other words, once we derive the sample weights, we can use them to sample high-quality subsets at different scales. The automatic reweighting process is much more efficient than introducing human revision. We will add more detailed explanations in the final version.
> >
> > **Data Generation.** We request the dataset of IMDB, SST-2 from the authors of  ZeroGen, and generate the data for the remaining tasks by ourselves. It takes 5 GPU hours for 200k synthetic data generation, which can then be used for downstream tasks. We’d like to mention that generating data from PLM is not the main focus of our work. Instead, we mainly focus on automatically learning the sample weights that are able to accurately measure the data quality. For completeness, we will add more background and detailed explanations about those works in our final version. Thank you for pointing this out!
> >
> >
> > Many thanks for your additional comments again! We would be happy to do any follow-up discussion or address any further comments. Looking forward to your further reply.
> >
> > Best regards,
> >
> > Authors

---

> ### Comment · Reviewer_23fs · 2022-12-11
> **Post-Rebuttal Updates**
>
> I'd like to thank the authors for providing very detailed responses and revisions to address my raised concerns in the original review. Please refer to my updated main review above (**Post-Rebuttal Updates**) for detailed comments.
>
> Overall, I'm happy with the paper revision, and I believe that it is a good paper with an important research goal, clear presentations, novel ideas, solid implementations, good theoretical insights, and promising empirical performance. I'll be happy to support the paper for acceptance. I've increased my rating from 5 to 8.

---

> > ### Author Response · Authors · 2022-12-12
> > **Response to Post-Rebuttal Updates**
> >
> > Dear Reviewer 23fs,
> >
> > We really appreciate your effort for reviewing our paper and your acknowledgement of our paper’s contribution. We are very glad that you liked the additional analyses w.r.t. the noise-robust loss function, the empirical results of RCE loss and the relaxed theoretical assumption. Our response to your further comments are as follows:
> > - We are very glad that you find the surface plots and training curves to be good empirical evidence, and we agree that a more principled and theoretically rigorous proof may be possible to support the assumption, which we will study in more detail in future work.
> > - We appreciate that you are happy with the more relaxed assumption (Assumption 3) in our updated version. Thank you again for pointing out this potential problem and helping us improve the quality of our paper!
> > - We are glad that you find our updated ablations for comparing different loss functions/noise-robust training strategies promising, adding those experiments indeed makes our paper more complete.
> > - Thank you for your advice! We will clarify more about the computational cost in the next version of our paper. By the quoted sentence in our previous response, we intended to say that we only need to optimize the weights once, after which the weights can be used as sample-wise probabilities. Given a fixed budget, we can directly sample high quality subsets from the entire dataset. Obtaining such a subset to train the TAM enables deriving similar or even better performance than the entire dataset, while saving the computational cost for training.
> >
> > Again, we thank you for acknowledging that our paper has an important research goal, the presentations are clear, the ideas are novel, the implementations are  solid, the theoretical analysis is insightful, and the empirical performance is promising.

---

### Official Review · Reviewer_h9eZ · 2022-10-23

**Confidence:** 2
**Clarity, Quality, Novelty And Reproducibility:** 1. Clarity. The paper is well-written…
**Correctness:** 3
**Technical Novelty And Significance:** 3
**Empirical Novelty And Significance:** 4
**Recommendation:** 8

**Strength And Weaknesses:**

Strength
1. The bi-level optimization idea is novel and natural, and the proof is easy to follow. By using the noise-free loss over learning the data sample weighting instead of optimizing the model directly, the search space is much smaller and thus easier to optimize.
2. The experiment results are sufficient and well-supported the effectiveness of this approach.
3. Overall the paper is clear and well-written and easy to follow.

Weaknesses:
1. In the ablation study, it would be good to provide some analysis between the proposed bi-level framework verses a single level baseline where the model is optimized using reversed cross-entropy directly.
2. In table 5, both ZEROGEN and SUNGEN's diversity is much smaller than Gold. It is unclear whether it is because of the model generated "hard" examples or noisy examples.



**Summary Of The Paper:**

The task this paper targeted is to train a model purely on a synthetic dataset without a clean oracle validation set. The paper proposed a bi-level algorithm where the coarse level uses a noise-robust loss optimizing re-weighting each data sample, and the fine level trains a model under the given weighted data sample dataset using regular cross-entropy loss. The noise-robust loss used is the reversed cross-entropy loss. The paper verified the effectiveness of this approach by both proved the convergency of this algorithm and empirically shows accuracy improvement over 8 text classification tasks.

**Summary Of The Review:**

The bi-level approach proposed in this paper is intuitive and is supported by good experimental results. The lack of analysis on the choice of noise-robust loss function and the lack of baseline for training directly with reversed cross entropy loss weakens the arguments of this paper.

---

> ### Author Response · Authors · 2022-11-15
> **Reply to Reviewer h9eZ**
>
> We sincerely thank Reviewer h9eZ  for the positive recommendation as well as the valuable suggestions. We really appreciate your kind words that our work is intuitive and has good results. Below we would like to give detailed responses to each of your comments.
>
> **Q1. “Analysis between the proposed bi-level framework verses a single level baseline.”**
>
> We show the comparison between our method and a single-level baseline in the following table. The results show that our method using RCE as the outer objective significantly surpasses the one-level optimization using RCE, which verifies the point of  (Wang et al., 2019; Zhang & Sabuncu, 2018) that using $l_\text{robust}$  to train the network parameters $\boldsymbol{\theta}$ leads to difficulty in optimization and hampers the network’s performance. The results also prove that adopting $l_\text{robust}$ as the objective in the outer loop implicitly overcomes the optimization difficulty of $l_\text{robust}$.
>
> | Method       |IMDb |Amazon |Yelp
> | ----              |----  | ---|---
> |RCE  one-level  |81.92  |81.50 |85.36
> |SunGen          |86.56  |84.63|90.38
>
> *(Appendix Table 11; Experiments run on 1,000k synthetic data )*
>
> **Q2. “ In table 5, both ZEROGEN and SUNGEN's diversity is much smaller than Gold. It is unclear whether it is because of the model generated "hard" examples or noisy examples.”**
>
> Thank you for pointing out this potential confusion.  The lower SelfBLUE4 score actually indicates higher diversity, and both ZeroGen and SunGen’s diversity are much higher than gold data thanks to the capacity of the large PLM. To avoid misunderstanding, we have highlighted the metric explanation in the caption of Table 5 in the current version.
>
> **Q3. “The lack of analysis on the choice of noise-robust loss function.”**
>
> Thanks for the great suggestion. We agree that it is necessary to add analysis on the choice of different noise-robust loss. We conduct extra experiments by replacing the RCE loss with Mean Absolute Error(MAE) loss (Ghosh et al. & Zhang and Sabuncu)  as the outer objective of SunGen. The experiment results show that the two losses demonstrate similar performances. We also have attached the results to Appendix A in the current version.
> |   Method        | IMDb |Amazon |Yelp
> |  ----           | ----  | ----  | ----
> |      ZeroGen    | 71.52 | 80.48 |84.95
> |     SunGen-rce  | 84.10 |84.22  |89.06
> |     SunGen-mae  | 84.23 |84.05  |89.23
>
> *( Experiments run on 200k synthetic data. )*
>
> Overall, we greatly appreciate your efforts for your thoughtful and insightful comments on our paper. We hope our answers have addressed your concerns.  We have revised the paper to clear the issues you mentioned in the comments in our latest version.
>
> **Reference:**
>
> [1] Wang, Y., Ma, X., Chen, Z., Luo, Y., Yi, J., & Bailey, J. (2019). Symmetric cross entropy for robust learning with noisy labels. In Proceedings of the IEEE/CVF International Conference on Computer Vision (pp. 322-330).
>
> [2]  Zhang, Z., & Sabuncu, M. (2018). Generalized cross entropy loss for training deep neural networks with noisy labels. Advances in neural information processing systems, 31.

---

### Official Review · Reviewer_1YN5 · 2022-10-26

**Confidence:** 4
**Correctness:** 4
**Technical Novelty And Significance:** 3
**Empirical Novelty And Significance:** 3
**Recommendation:** 8

**Clarity, Quality, Novelty And Reproducibility:**

The paper is clearly written. The method is novel by proposing a self-guided data generation framework that is robust to noisy labels while requiring no human annotations. The results should be largely reproducible with their provided code and their plan for open-sourcing.

**Strength And Weaknesses:**

Strengths:
1. The research problem of avoiding noisy samples in the new paradigm of data-generation-based zero-shot setting is well-motivated, and the underlying assumption of this work that there is no human annotation available for validation looks more realistic though challenging.
2. The proposed method looks pretty convincing both theoretically and empirically. The experiment performance is very strong compared to several fair baselines.
3. Further analysis especially the visualization in Figures 1 and 3 provides clear evidence for the hypothesis that the proposed method is effective for filtering out noisy labels.
4. The paper is clear and well-written.

Weaknesses:
I didn't see significant weaknesses in this paper such that it should not be accepted.


**Summary Of The Paper:**

This paper proposes SUNGEN, a framework to automatically construct high-quality data for zero-shot classification problems. The method first employs pretrained language models to generate samples for a given task and then learns the sample weights indicating data quality without requiring any human annotation. The paper provides theoretical proof that the proposed method can be noise-free. Moreover, experiments on eight text classification tasks demonstrate the strong performance of the proposed method and its ability to identify and ignore noisy samples.

**Summary Of The Review:**

The paper proposes a novel method to automatically avoid noisy synthetic data in the paradigm of data-generation-based zero-shot learning. Clear analysis is provided both theoretically and empirically to show the effectiveness of the proposed method. The paper is worth acceptance.

---

> ### Author Response · Authors · 2022-11-15
> **Reply to Reviewer 1YN5**
>
> We sincerely thank Reviewer 1YN5  for the positive feedback and we are grateful for the time you spent on our submission. We are also glad for the acknowledgment that the problem we are working on is realistic and that the method we propose is well-motivated. We hope our paper can provide contributions to further exploring the direction of data-generation-based zero-shot learning via PLM.

---

### Official Review · Reviewer_mEap · 2022-10-26

**Confidence:** 3
**Correctness:** 3
**Technical Novelty And Significance:** 3
**Empirical Novelty And Significance:** 3
**Recommendation:** 5

**Clarity, Quality, Novelty And Reproducibility:**

The paper appears to be clear-enough and technical details are mostly sufficient. However, it is not clear how the hyper-parameters were tuned, in the absence of clean validation sets.

**Strength And Weaknesses:**

Positives:
- The experimental results look very promising with significant improvements over sensible baselines
- The approach overall is rational and makes sense
- The theoretical analysis is a plus (though does not seem to give strong insights)

Negatives:
- Some claims in the paper are not fully convincing, e.g. "In Sec. 4, we theoretically show that using Lrobust as the outer objective, our method can find a set of sample weights w∗ with just the synthetic validation set, such that w∗ maximizes the model performance on the clean data." This appears to be valid only for certain types of noises, one can easily come up with structured noise cases (as common in reality) where the weighting can easily attend to noisy samples. In particular, if the model generated samples are consistently problematic (rather than having uniform noise or being noisy only in a subset of samples), then the theoretical proof would not apply.
- "Since lrobust is now used to optimize the sample weights w, which is in a much smaller search space and has simpler structure than θ, thus is easier to optimize." Here, I get the intuition, however, I am not sure if it is definitely correct. In particular, one can see sample weights as a continuous relaxation to the sample selection problem. As sample selection is a combinatorial problem, it is likely to be a much harder optimization problem, compared to the naturally-continuous model parameter optimization problem (despite having a lot more optimization variables and similarly being non-convex).
- The method is evaluated as a whole bundle with PLM. No baselines are presented as alternatives to various model components. I couldn't see a detailed model ablation study, and I wonder about alternative choices in certain parts of the model, e.g. what happens if we were to use a different robust loss or just xent loss in the outer loop?


**Summary Of The Paper:**

The paper focuses on the problem of utilizing pretrained large-scale language models for training data generation (based on careful prompts), to be used in training zero-shot classifiers. This approach stands out as an alternative to fine-tuning the language model for a new task (again for data generation) or direct prompt-based zero-shot classification.

The paper mainly aims to tackle the problem of managing sample noise in the training data generated by the language model, in an automatic way (avoiding human intervention). For this purpose a bi-level optimization approach (meta-learning-like) is proposed, where the outer loop estimates the sample training weights based on a robust loss function and the inner loop trains the classification model based on the given sample weights.

**Summary Of The Review:**

Overall, the paper appears to be promising with strong positive results. However, it lacks in certain model and experimental aspects as discussed above. In particular, in my preliminary rating, I find the the lack of clarity regarding model selection and the lack of experiments evaluating major model choices (in particular on the effects of robust loss) as major experimental limitations.

---

> ### Author Response · Authors · 2022-11-15
> **Reply to Reviewer mEap (Part 1/2 )**
>
> We sincerely thank Reviewer mEap for your review and are grateful for the time you spent on our submission. We are also glad you think our paper is promising and obtain strong positive results.  Below we would like to give detailed responses to each of your comments.
>
> **Q1 “Some claims in the paper are not fully convincing. This appears to be valid only for certain types of noises, one can easily come up with structured noise cases (as common in reality) where the weighting can easily attend to noisy samples. In particular, if the model-generated samples are consistently problematic (rather than having uniform noise or being noisy only in a subset of samples), then the theoretical proof would not apply.”**
>
> Thank you for your constructive comments. We do rely on certain assumptions for the theoretical property, which are located in Appendix C.2. To avoid confusion, we have modified the claims accordingly in the revised version (e.g. "In Sec. 4, we theoretically show that using $L_\text{robust}$ as the outer objective, our method can find a set of sample weights $w^∗$ with just the synthetic validation set" >  “ In Sec. 4, under the condition that the majority of data are correctly labeled (Ghosh et al., 2017; Wang et al., 2019), we theoretically show that using $L_\text{robust}$  as the outer objective, our method can find a set of sample weights $w^∗$ with just the synthetic validation set” ).
>
> For deriving theoretical guarantees, we refer to the common assumptions in the noisy label learning literature (Wang et al., 2019; Ghosh et al., 2017; Zhang and Sabuncu, 2018), since the structured noise cases are not properly defined in theory. It is worth mentioning that the noise contained in the generated data is induced by the PLM itself, rather than artificially designed. The superior experimental results on the practical scenario demonstrate that our SunGen is able to perform well in reality.
>
> **Q2 "One can see sample weights as a continuous relaxation to the sample selection problem. As sample selection is a combinatorial problem, it is likely to be a much harder optimization problem.”**
>
> Thank you for making a great point. Sampling a discrete subset is a difficult combinatorial problem. However, we aim to solve the relaxed problem that finds a set of continuous sample weights. Note that reweighting is an approximation of the original sample selection task that achieves a similar effect, which enables us to leverage gradient information to conduct optimization and thus is easier to solve. This continuous relaxation technique is prevalent and proven useful in previous literature (Borsos, Z. et al; Ren, M. et al; Zhou, X. et al).
>
> **Q3 “I wonder about alternative choices in certain parts of the model, e.g. what happens if we were to use a different robust loss or just xent loss in the outer loop”**
>
> Thank you for the suggestion. As suggested, we add the experiments using different robust loss and CE loss as the outer objective of SunGen. The results are listed in the following table.  The results show that different noise robust losses (e.g. RCE, Mean Absolute Error(MAE) loss) in SunGen achieve similar performance. These noise robust losses significantly surpass the bi-level framework using CE loss as the outer objective.  Due to the space limit, we put the model ablation results in Appendix A.
> | Method        |  IMDb |Amazon |Yelp
> |  ----              | ----  | ----  | ----
> |      ZeroGen    | 71.52 | 80.48 |84.95
> |     SunGen-ce  | 74.05 | 79.81 |83.90
> |     SunGen-rce  | 84.10 |84.22  |89.06
> |     SunGen-mae  | 84.23 |84.05  |89.23
> *( Experiments on 20w synthetic data)*
>
> **Q4  “how the hyper-parameters were tuned, in the absence of clean validation sets”**
>
>
> Due to the absence of clean validation sets in the zero-shot setting, we keep the TAM model training hyper-parameters (e.g. batch size, training epoch, learning rate) the same across different tasks following (Ye et al, 2022).  For selecting the appropriate value of learning rate in the outer loop, we select from {2.5e-1, 1e-1, 1e-2} according to the value of RCE loss in the outer loop. If the outer loss steadily decreases and reaches a low value, then it indicates that the optimization is going well.  We have attached the implementation details in Appendix F.2. Furthermore, we have submitted our code and will make it public upon publication. With the released codes and implementation details, it should be easy to reproduce our results.
>
>
> Thank you very much for the constructive comments, which really help us further improve our work. We hope our answers have addressed your concerns. If you have any further questions, we are happy to address them.

---

> > ### Author Response · Authors · 2022-11-15
> > **Reply to Reviewer mEap (Part 2/2 )**
> >
> >
> >
> > **Reference:**
> >
> > [1] Borsos, Z., Mutny, M., & Krause, A. (2020). Coresets via bilevel optimization for continual learning and streaming. Advances in Neural Information Processing Systems, 33, 14879-14890.
> >
> > [2]  Ren, M., Zeng, W., Yang, B., & Urtasun, R. (2018, July). Learning to reweight examples for robust deep learning. In International conference on machine learning (pp. 4334-4343). PMLR.
> >
> > [3] Zhou, X., Pi, R., Zhang, W., Lin, Y., Chen, Z., & Zhang, T. (2022, June). Probabilistic Bilevel Coreset Selection. In International Conference on Machine Learning (pp. 27287-27302). PMLR.
> >
> > [4] Wang, Y., Ma, X., Chen, Z., Luo, Y., Yi, J., & Bailey, J. (2019). Symmetric cross entropy for robust learning with noisy labels. In Proceedings of the IEEE/CVF International Conference on Computer Vision (pp. 322-330).
> >
> > [5] Ghosh, A., Kumar, H., & Sastry, P. S. (2017, February). Robust loss functions under label noise for deep neural networks. In Proceedings of the AAAI conference on artificial intelligence (Vol. 31, No. 1).
> >
> > [6] Zhang, Z., & Sabuncu, M. (2018). Generalized cross entropy loss for training deep neural networks with noisy labels. Advances in neural information processing systems, 31.
> >
> > [7] Ye J, Gao J, Li Q, et al. Zerogen: Efficient zero-shot learning via dataset generation. arXiv preprint arXiv:2202.07922.

---

> > > ### Author Response · Authors · 2022-11-17
> > > **Further comments and discussions will be appreciated!**
> > >
> > > Dear Reviewer mEap,
> > >
> > > Thank you for your valuable time to review our work and for your constructive feedback. We posted our response to your comments two days ago, and we wonder if you could kindly share some of your thoughts so we can keep the discussion rolling to address your concern if there are any.
> > >
> > > In the previous response,
> > >
> > > 1. As suggested, we state the assumptions and modified the claims accordingly in the revised version. It is worth mentioning that even with a simplified noise assumption for deriving theoretical guarantees, the superior experimental results on the practical scenario demonstrate that our SunGen is able to perform well in reality, rather than artificially designed noise.
> > > 2. We answer why sample weight relaxation is not a harder optimization problem. The main idea is that reweighting is an approximation of the original sample selection task that achieves a similar effect, which enables us to leverage gradient information to conduct optimization and thus is easier to solve.
> > > 3. We add the experiments using different robust loss and CE loss as the outer objective of SunGen. The results show that different noise robust losses in SunGen achieve similar performance. These noise robust losses significantly surpass the bi-level framework using CE loss as the outer objective.
> > > 4. We state our hyper-parameter selection method:  we keep the TAM model training hyper-parameters the same across different tasks following (Ye et al, 2022).  For selecting the appropriate value of learning rate in the outer loop, we select according to the value of RCE loss in the outer loop.
> > >
> > > We would appreciate it if you could kindly take a look at both the revision and our response to your comments. If you have any further questions, we are happy to discuss them!
> > >
> > > Best regards,
> > >
> > > Authors

---

> > > > ### Author Response · Authors · 2022-11-28
> > > > **Follow up to reveiwer mEap**
> > > >
> > > > Dear Reveiwer mEap,
> > > >
> > > > We would like to thank you again for your detailed reviews. We have updated our draft and added replies to your three Cons with our latest experimental results.
> > > >
> > > > Since the rebuttal deadline is approaching soon, a lot of papers have finished the discussion.  Given that your current score is 5, we would appreciate it if you could let us know if our responses have addressed your concerns satisfactorily. If your concerns have not been resolved, could you please let us know about it so that we have the opportunity to respond before the deadline?
> > > >
> > > > We would be happy to have any follow-up discussions or address any additional concerns.
> > > >
> > > > Thanks very much! Looking forward to your reply.
> > > >
> > > > Paper2800 Autors

---

### Decision · Program_Chairs · 2023-01-20

**Decision:**

Accept: notable-top-25%

**Justification For Why Not Higher Score:**

The noise-free claim is too strong. Theoretical justification doesn't support that claim.

**Justification For Why Not Lower Score:**

Good paper overall. It definitely offer more insights into zero-shot learning based on large pretrained models.

**Metareview: Summary, Strengths And Weaknesses:**

Generating task-specific samples from a pretrained VLM significantly alleviates the annotation cost of downstream tasks. This paper proposed a new sample generation method that is less noisier than existing methods. With experimental and theoretical justifications, the proposed method (SunGen) has good potential in zero-shot text classification tasks.

Strength:
1. A new framework to automatically construct high-quality data from pretrained VLMs for zero-shot classification problems. The proposed one is clearly different from the existing ones.
2. Comprehensive experiments plus theoretical analysis.
3. Well-written presentation.

Weakness:
1. The theoretical proof is not very insightful. It does not answer why the proposed method is fundamentally noisy-free but other methods are not. The theoretical part could be moved to Appendix.

**Note From Pc:**

if the above contains the word "oral" or "spotlight" please see: "oral" presentation means -> notable-top-5% and "spotlight" means -> notable-top-25%. As stated in our emails, we are disassociating presentation type from AC recommendations